# Development of "Physical Parametrizations with PYthon" (PPPY, version 1.1), and its usage to reduce the time-step dependency in a microphysical scheme

Sébastien Riette[1]

[1]CNRM, Université de Toulouse, Météo-France, CNRS, Toulouse, France

**Correspondence:** Sébastien RIETTE (sebastien.riette@meteo.fr)

**Abstract.**

To help develop and compare physical parametrizations such as those found in a numerical weather or climate model, a new tool was developed. This tool provides a framework with which to plug external parametrizations, run them in an offline mode (using one of the two time-advance methods available), save the results and plot diagnostics. The software can be used in an 0D and a 1D mode with schemes originating from various models. As for now, microphysical schemes from the Meso-NH model, the AROME (Applications of Research to Operations at Meso-scale) model and the Weather Research and Forecasting model have been successfully plugged. As an application, PPPY is used in this paper to suppress the origin of the time-step dependency of the microphysical scheme used in the Météo-France small scale operational numerical weather model. The tool helped to identify the origin of the dependency and to check the efficiency of the introduced corrections.

## 1   Introduction

A weather or climate numerical model contains several parametrizations (e.g. turbulence, convection . . . ) which interact not only together but as well with the dynamical core. When a parametrization is being developed or debugged, these interactions can hide and/or amplify a tested modification through compensatory errors and feedbacks. When the goal is to compare two parametrizations hosted by different models, these interactions distort the comparison as the other model components can be very different (dynamical core, discretization and other parametrizations). To circumvent these effects, one can reduce the interactions by unplugging other parametrizations (ideal cases, aqua-planet experiments) or by reducing the problem size (2D vertical simulations, single column model).

The choice of the comparison strategy depends on the intended goal: a full 3D-model is able to represent all the interactions whereas simplified models represent only a subset of these interactions. Even the single column model however (one of the simplest configurations) is not always sufficient for separating the impact of the different parametrizations (see, for example,

point 8 of Ghan et al. (2000) conclusion). A simpler framework could therefore be useful; this can be a toy model in which only one parametrization is plugged and compared to other parametrizations plugged in the same way.

An example of a toy model used to compare microphysical schemes is given by Shipway and Hill (2012) using the Kinematic Driver (KiD) model. This is also the approach taken here to develop PPPY in which one can plug existing parametrizations from different models, deal with the simulations, compare the outputs and plot the results. The tool described here has some common points with the KiD model but is able to deal with any parametrization (not only microphysics), integrates the graphical part and is very flexible through the use of the Python language to control the execution flow (for example running and comparing hundreds of different configurations is not an issue and incorporating new diagnostics is simple). The KiD model, for its part, allows advection and, hence, lies between PPPY and a Single Column Model.

Development motivation came from the observation of a difference on 3D simulations with the Meso-NH model (Lac et al., 2018) when the time step was changed. The impact of the time step being greatest for rain accumulation and when prognostic hail was activated, the one-moment microphysical scheme (Pinty and Jabouille, 1998, hereafter referenced as ICE) was suspected to be the prime reason. To assess this dependency, simulations in a 0D mode, using only the core microphysical processes (collections, riming, vapor deposition, evaporation, . . . ), excluding the saturation adjustment and the sedimentation, are performed using PPPY. To test all the processes, an initial state involving all the hydrometeors was chosen. The initial mixing ratios were quite large ($10 \, \mathrm{g \, kg^{-1}}$ for vapor, no hail and $1 \, \mathrm{g \, kg^{-1}}$ for the other hydrometeors) and hail was activated (even if the illustrations here were made with simulations without hail to simplify the plots); the initial temperature was set to 270 K. The setup is not fully realistic (with an unrealistically high supersaturation) but allows simulations to involve all the species and, hence, virtually all the microphysical processes. It was checked (not shown) that the time-step dependency still exists when using realistic initial values for the relative humidity.

When several hydrometeors are mixed in a model cell without exchange of matter or heat with the exterior, the microphysical processes tend towards a state of equilibrium. This final state must not depend on the time step used. In addition, when two (or more) simulations running with different time steps are compared, they should have the same results for common output times. In Fig. 1, we can see that the final state depends on the time step used (between 1 s and 60 s) for water vapor, rain and temperature. The chaotic appearance of this plot is furthermore a signature of the time-step dependency. Without this drawback, all curves of a same color would normally follow the same time evolution. For example, after 60 s of simulation, great uncertainty exists on the hydrometeors presence; depending on the time step used, rain, graupel and snow may (with very significant content) or may not exist. It should be noted here that 60 s is the order of magnitude of the time-step length used in the Météo-France small scale operational numerical weather model, AROME (Application of Research to Operations at Mesoscale, Seity et al., 2011), which shares the same physical package with the Meso-NH model.

In the COnsortium for Small-scale MOdeling (COSMO) model, Barrett et al. (2019) also observed a time-step dependency on rain and hail accumulations traced back mainly to the interaction between the dynamics and the physics of the model, and, to a lesser extent, to some microphysical processes. The example shown in their paper demonstrates that a significant part of the time-step dependency can also be explained by the microphysical scheme itself.

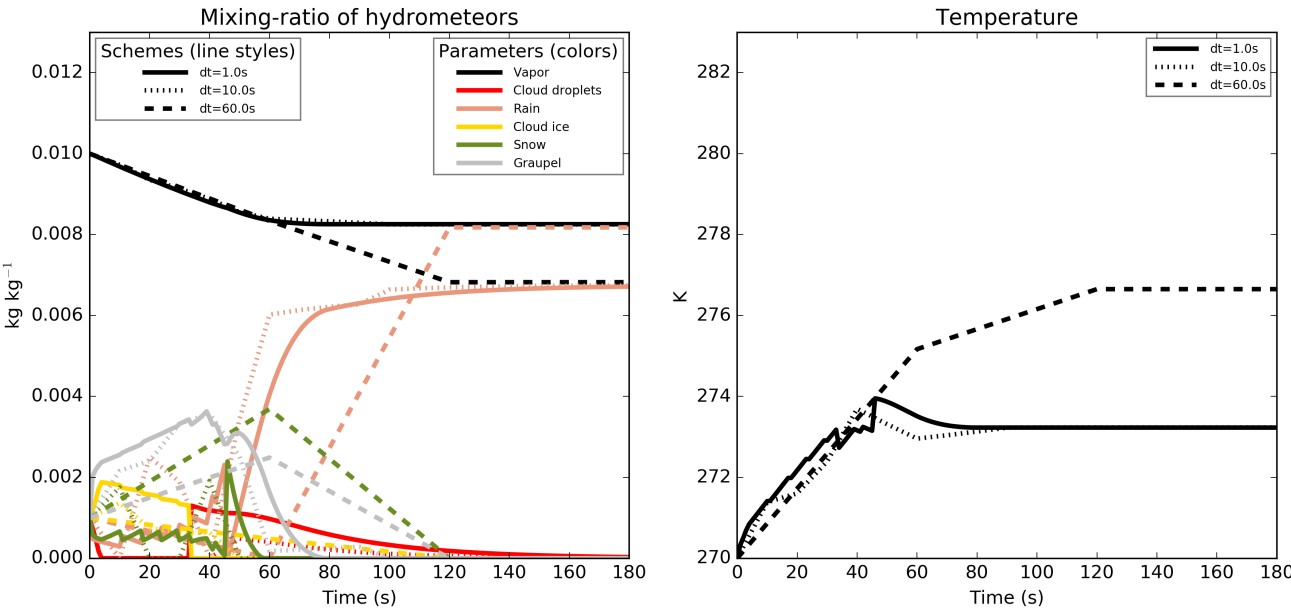

**Figure 1.** Time evolution of the mixing ratio of the different hydrometeors (in kg/kg, left panel) and of the temperature (in K, right panel). The simulations were performed using time steps between 1 s and 60 s.

The time-step dependency of the microphysical scheme is not specific to the ICE parametrization. The dependency is also observed with the Liquid Ice Multiple Aerosols (LIMA) scheme (Vié et al., 2016) (a quasi two-moment microphysical scheme in development in the Meso-NH and AROME models, which is built upon the ICE one-moment scheme). In the simulations performed with this scheme (Fig. 2), the setup is the same as for the ICE scheme but the saturation adjustment is active.

Such a dependency was also observed in the Integrated Forecasting System (IFS) model (Forbes, 2018), and related to the formulation of the warm-rain processes. Moreover, some microphysical schemes of the Weather Research and Forecasting (WRF) Model (version 3.9.1.1) have been plugged and also exhibit time-step dependency, as shown in Fig. 3 for the Eta (Ferrier) scheme (Rogers et al., 2001, panel a), the Milbrandt–Yau Double Moment scheme (Milbrandt and Yau, 2005a, b, panel b), the Morrison 2–moment scheme (Morrison et al., 2009, panel c), the Hebrew University of Jerusalem Spectral Bin

Microphysical (HUJI SBM) scheme (Khain et al., 2004, panel d), the Thompson scheme (Thompson et al., 2008, panel e) and the WRF Single–moment 6–class (WSM6) scheme (Hong and Lim, 2006, panel f). The WRF simulations are performed using the saturation adjustment included inside each scheme. These examples also show how PPPY can be useful for the exhibiting of some of the behaviors of a scheme (such as time-step dependency, oscillations and water conservation) independently of the other model components.

Section 2 describes the technical choices and provides an overview of what can be done with the software. Some examples of usage are given in Sect. 3 before the conclusion.

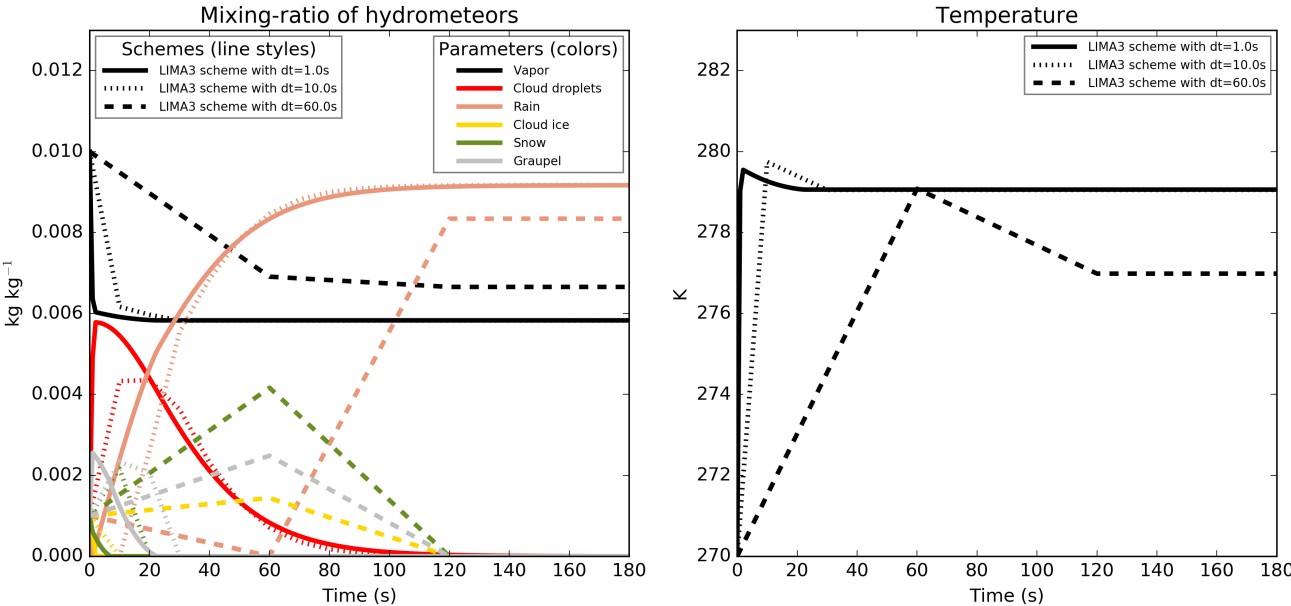

**Figure 2.** Same as Fig. 1 but using the LIMA scheme.

## 2 Functionalities and technical aspect

Documentation is provided with the software (see the code availability section). To complement this documentation, this section gives some details on how to add a parametrization, how the software works as well as describing the functionalities related to the parametrizations and the comparisons.

### 5 2.1 Technical aspects

The tool consists of a Python package which drives the simulations and performs the comparison: initialization, the calling of the Fortran routines (using the original source code of the parametrization), the saving of the results (in HDF5 files using the h5py module) and the plotting of the results (through the matplotlib module).

Two kinds of objects exist: those which represent a parametrization, and those representing the comparison. A standard object (an abstract class) is provided in order to define a parametrization (the PPPY box in Fig. 4). This abstract class already contains everything needed to perform the time advancing and the saving of results but must be complemented (by inheritance) to incorporate the actual call to the different parametrization codes (Param1 and Param2 boxes of the figure). Finally, each parametrization can be used with different configurations. To achieve this, different instances (Param1.1, Param2.1 and Param2.2 boxes) are created, one for each of the configurations (e.g. time-step length, options specific to the parametrization).

For the comparison, the provided class (PPPYComp in the figure) can be used directly or can be complemented (by inheritance, UserComp box in the figure) to add new diagnostics (e.g. new plot kind, computation of a derived variable to plot).

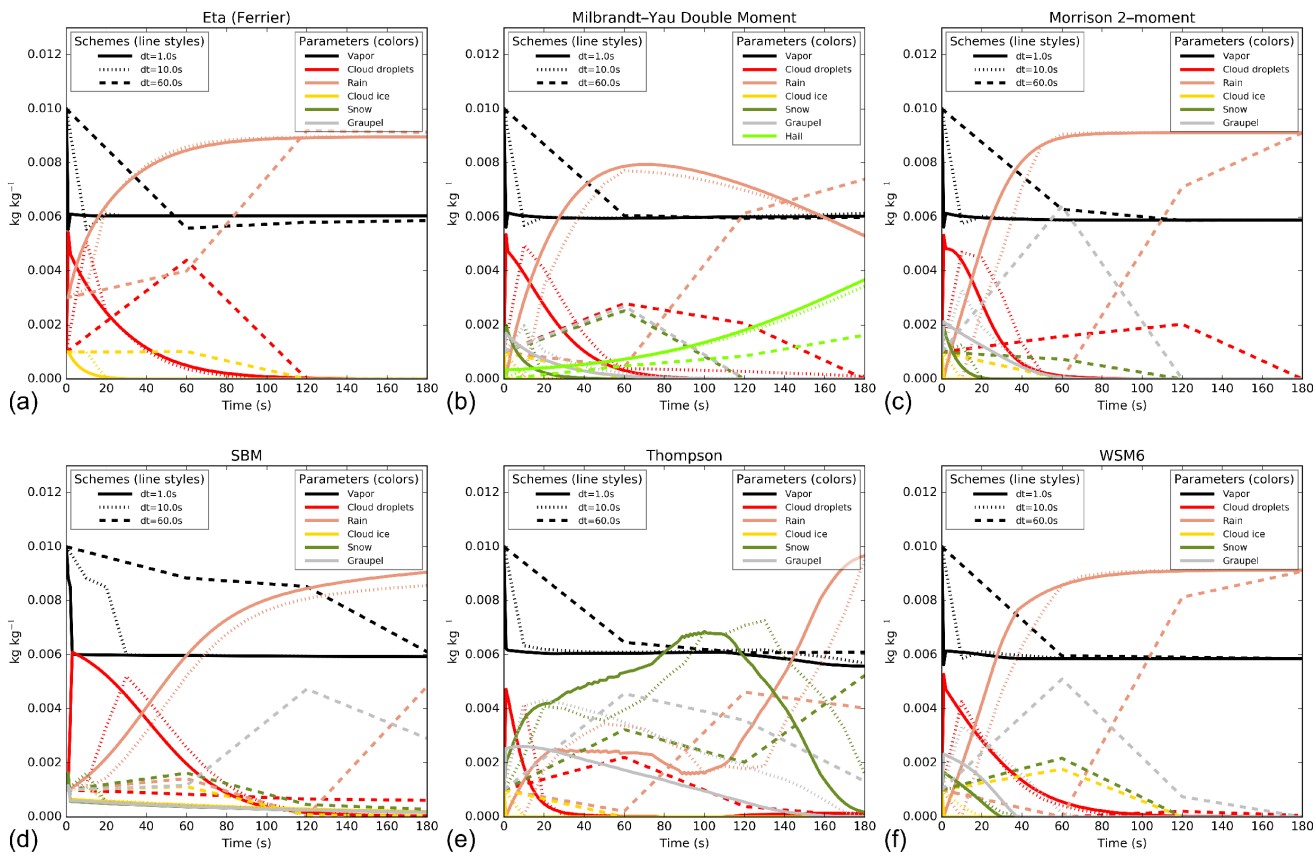

**Figure 3.** Same as left panel of Fig. 1 but using some schemes of the WRF model identified in the panel titles (see text for complete references).

An instance of the class is created for each comparison to perform (Comp box). A comparison is defined by the list of the parametrizations to use, the simulation length and the initial state. This comparison instance drives the parametrization instances to carry out the simulations and to plot the result.

These different objects are described in the following subsections. In addition, in the provided package, the `examples` directory contains, among other items, a test example which is commented on in Appendix A so as to illustrate the different steps described in the current section.

### 2.1.1 The low level part of the parametrization: source code and compilation

The trickiest part comes from the interfacing between Python and the parametrization. This part is quite technical but important as the main difficulty in using PPPY with a new parametrization lies in this interfacing task.

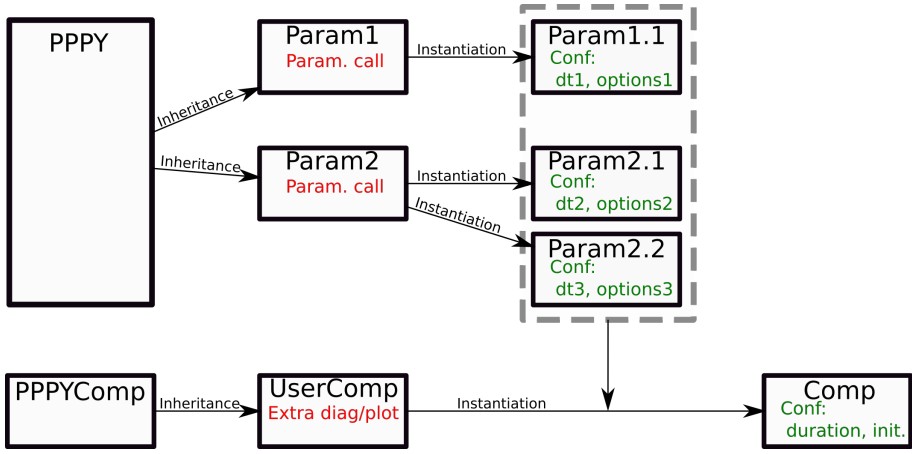

**Figure 4.** Tool organisation diagram. The 6 top boxes represent the parametrization objects while the 3 lower boxes are the comparison objects.

If the parametrization was written using Python (like the box-Lagrangian scheme used in Sect. 3.2) the interfacing would be straightforward. But, numerical weather and climate models often use Fortran and a module is needed to perform the interfacing with Python. There are several ways in which to accomplish this task; this paper does not set out to review these ways but, here, are listed the two used at some point in the development process. These two ways can still be used even if the examples

provided with the package use only the second one. Both methods aim at building a library (a collection of binary codes) which contain all the Fortran codes needed to call a given parametrization. The f2py utility can be used; it helps to build a shared library suitable to be imported and used from Python but it can be difficult to use (in particular the built library depends on the exact Python version and the argument order is not always preserved). The second one is a module (named ctypesForFortran) provided with PPPY which acts directly on a shared library built from the Fortran source code. The interfacing consists of

defining the signature (the interface of the subroutine written in a specific way) of each Fortran subroutine or function to employ.

The PPPY user is free to use whichever Python-Fortran interfacing method (among the two aforementioned or other ones). The ctypesForFortran way intends to help the interfacing of Fortran functions and subroutines on a Linux system. It handles memory allocations and array memory order. Internally ctypesForFortran uses the Python ctypes module (which normally

handles the C shared libraries) to interact with the library without adding a C or Fortran layer. It deals with Boolean, strings, integers and floats (32- and 64-bits) but does not support structures. The array and string arguments must be explicitly defined (no ":", ".." or "*" are allowed in the interfaces) and no argument can be optional. If this is not the case, a wrapper must be written in Fortran meeting these requirements and calling for the original subroutine.

For the potential C-written parametrizations, interfacing can directly employ the ctypes module.

The compilation being a complex process (that can involve scripts that modify, on the fly, source codes), it could be difficult to isolate and compile, outside of the box, the source code needed for a given parametrization. To reduce this difficulty, the various examples provided with PPPY (in the example directory) follow this procedure:

- Modification of the model compilation script and/or Makefile file to include the option to build a position-independent code, suitable for dynamic linking ("-fPIC" option),

- Normal compilation of the model,

- Use of the various object codes and/or static libraries built during the normal compilation step to build a shared library with the different entry points needed by the parametrization.

At this stage the remaining difficulty is to identify the different routines that must be called upon to perform the parametrization initialization and execution.

### 2.1.2 The high level part of the parametrization: the PPPY Python object

Once the compilation part is completed, a Python object must be created in order to manipulate the compiled library. An abstract class (PPPY) is provided for this purpose and must be used, by inheritance, to build a class specific to the parametrization employed.

The abstract class has placeholders for the requested standardized methods and these must be implemented.

Following the order of the execution flow, the first method to adapt is that of initialization. In this method, all the available options of the scheme are defined and consistency checks can be achieved. Among these options, one is the time step (mandatory); the others being specific to the scheme.

The following method concerns the setup. Here, the computations that need to be done once per simulation are performed. For example, it could be constant definitions, pre-calculation of lookup tables or files fetching.

The initial state provided at the beginning of a simulation is the same for all the parametrizations involved in the comparison. This state contains all the variables that must be monitored by all the schemes although it is common for a given scheme to need additional prognostic variables. For example, when comparing a one-moment microphysical scheme with a two-moments microphysical scheme, only the first moment (mass) can be compared but the two-moment scheme needs to monitor another moment (concentration). The build_init_state method is the place in which to define and initialize these additional variables and to add them to the initial state of the simulation. The output of this step is the first state saved in the output file.

The execute method is in charge of calling the actual code of the parametrization, making use of the compiled library. It might be necessary to perform numerical and/or physical conversions before and after this call. Indeed, the same quantities must be monitored by every scheme even if, internally, each scheme uses its own set of variables; physical conversion (for example changing the temperature variable from potential temperature to true temperature) may therefore be needed. Moreover, modifications in memory representation may also be required (all variables are 64-bit in the Python script but can be converted into 32-bit for instance).

This class is then instantiated by providing the required options (time step and options specific to the parametrization).

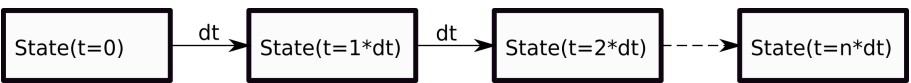

**Figure 5.** Time advance for a step-by-step simulation.

### 2.1.3 Comparison Python object

The parametrizations, which are instances created from the PPPY class (as stated above), are intended to be used by a Python object in order to perform a comparison. This comparison object (instantiated from the PPPYComp class, to get the Comp object of Fig. 4) is characterized by the list of parametrizations to use, the simulation length and the initial state of the simu-
lations. The comparison object is then responsible for the running of the different parametrizations (isolated from each other), the computing of diagnostics (which can be added by creating a custom comparison class by inheritance) and for the plotting of the results (plot methods can also be added).

### 2.2 Tool functionalities

PPPY allows the comparison of several parametrizations. The different parametrizations can differ from the underlying code
or can differentiate themselves by the choice of the parameters controlling the scheme. The comparison between two identical parametrizations using different time steps or different time-advance methods is also possible. Two time-advance methods exist:

**"step-by-step"** like a true simulation, the output is computed from the output of the previous time step (Fig. 5).

**"one-step"** the output (at all output times) is computed by a direct integration from the initial state (Fig. 6).

The development was conducted in such a way as to allow the comparison of any parametrizations, not only microphysical schemes. The set of variables monitored is not limited to predefined ones; the user can add any variable of any dimensions. Moreover PPPY is able to use schemes from different models (interfacing with AROME, Meso-NH and WRF parametrizations has been done).

Two plot kinds are currently available but others can be written by extending the tool. Plot methods already in existance
can draw results for 0D and 1D simulations. The y-axis is used to represent the value of the variables in the 0D simulations or the different points in the 1D simulations. The two plot kinds differ in the x-axis used to represent the time (with different plots superimposed for the different schemes, such as in the examples of Sect. 3.1) or the different schemes (with different plots superimposed for the different output times, as in Fig. 11 to 13 of Sect. 3.2). In this context, the different schemes can be different parametrizations and/or the same parametrization but using different options (constants, configuration options or
a time-step choice); this allows the performance of sensitivity tests to one parameter. The figures which illustrate the examples shown in this paper have been directly produced by the software.

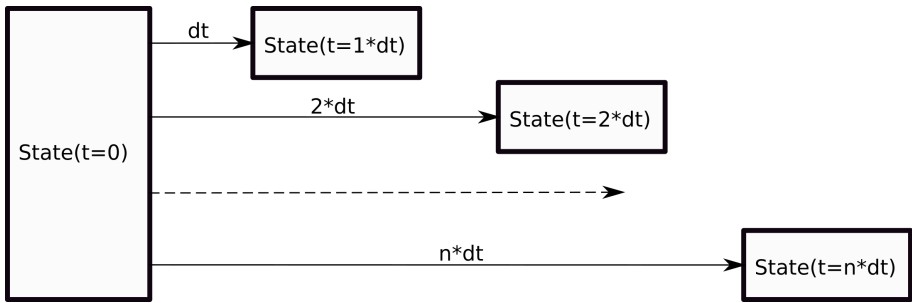

**Figure 6.** Time advance for a one-step simulation.

## 3 Application to microphysical parametrizations

The ICE microphysical scheme is divided into three parts: a statistical adjustment to the saturation (to balance cloud water and ice with the vapor, according to the temperature), the core microphysical processes and the sedimentation. Each of these three parts may contain sources of time-step dependency.

The adjustment to saturation modifies the temperature and hence modifies also the saturation point and then the cloud content. This feedback could be a source of time-step dependency but it was checked (not shown) that the saturation adjustment used reaches an equilibrium very quickly; the impact of a second iteration can barely be detected. The cloud ice fraction (the ice content divided by the total -ice and liquid- content) depends on the temperature (for temperatures above 0 °C, the cloud is liquid, for temperatures under -20 °C, the cloud is icy and the cloud ice fraction is linearly interpolated between these

two points) and a consumption of one of these two species in the core microphysical processes implies a consumption of the other species during the following saturation adjustment (to keep the cloud ice fraction consistent with the temperature). This mechanism leads to a time-step dependency, on which work will have to be carried out.

    In this section, two examples of the tool usage are shown. The first one deals with the time-step dependency due to the processes of the microphysical scheme (without adjustment and without sedimentation) in a 0D mode; the second one is a

comparison of several sedimentation schemes, in a 1D mode.

### 3.1 Time-step dependency in the microphysical scheme

The final goal of this application was to suppress the time-step dependency of the simulations shown in Fig. 1. To achieve this result, a new set of simulations was performed using much smaller time steps (between 0.001 s and 1 s) in order to look for a convergence between the simulations when the time step decreases. In Fig. 7, the final values are far more consistent but some

oscillations are still visible on the time evolutions and the values after an integration time of between 40 s and 60 s are still very uncertain. It was necessary to use very small time steps in order to obtain numerical oscillations smaller than the physical variations.

    The simulations have been carried out several times activating and deactivating the different microphysical processes. To do this, the processes have been called individually by the PPPY software (when they are written in separate subroutines)

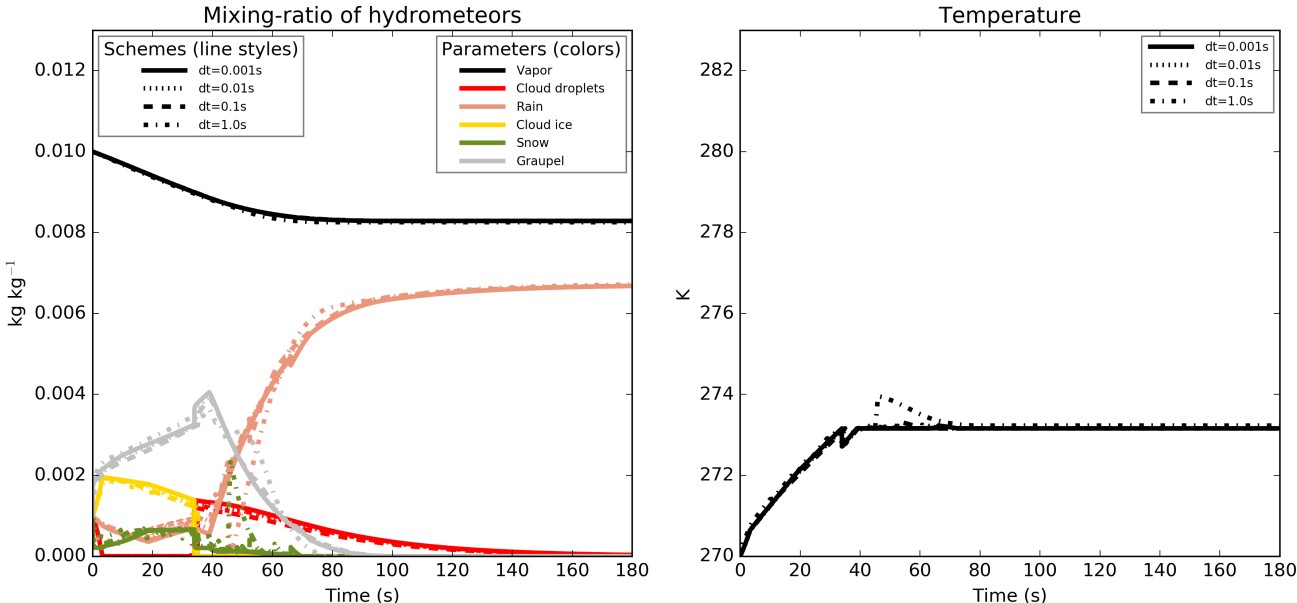

**Figure 7.** Same as Fig. 1 but using smaller time steps between 0.001 s and 1 s.

or activated through switches or, at worst, (un-)commented in the source code. This trial-and-error experimentation makes it possible to identify the processes that led to the oscillations and to the time-step dependency, and allowed the checking of each correction individually from the others.

The purpose of this paper is not to detail the modifications that were needed to suppress the time-step dependency, however 5   the more important ones are listed in Appendix B.

With the revised version of the microphysical scheme, the simulated values for the different simulations shown in Fig. 7 and Fig. 1 are now perfectly indistinguishable in Fig. 9 and Fig. 8 for every common output times (the simulation with a 60 s time step, with the dashed line in the figure, provides outputs only after 60, 120 and 180 s of integration time).

The 0D simulations were very useful for the identifying and correcting of the processes involved in the time-step dependency 10   of the ICE microphysical scheme. It would certainly have been possible to achieve the same result with another method but this one was convenient (0D simulations are very rapid and a single tool performs the simulations, compares the outputs and plots the comparison) and allowed the complete isolation of the processes of interest from the other parts of the model (dynamics with transport, other physical parametrizations, sedimentation and adjustment).

### 3.2   Sedimentation schemes

15   A similar time-step sensitivity test is done regarding the sedimentation scheme used in the model (all other parametrizations, including the microphysical processes, were turned off). Two schemes are available: the AROME operational one (Bouteloup et al. (2010), BSB2010 hereafter), which is a statistical scheme and the Eulerian scheme included in the original version of

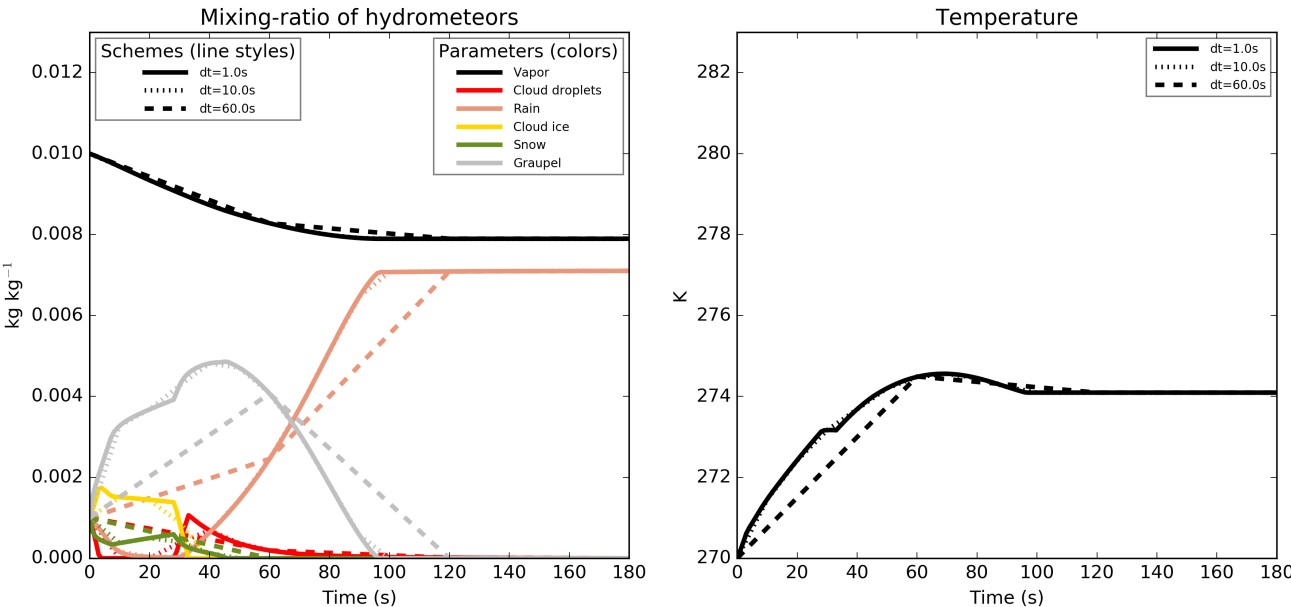

**Figure 8.** Same as Fig. 1 but using the new version of the microphysical scheme.

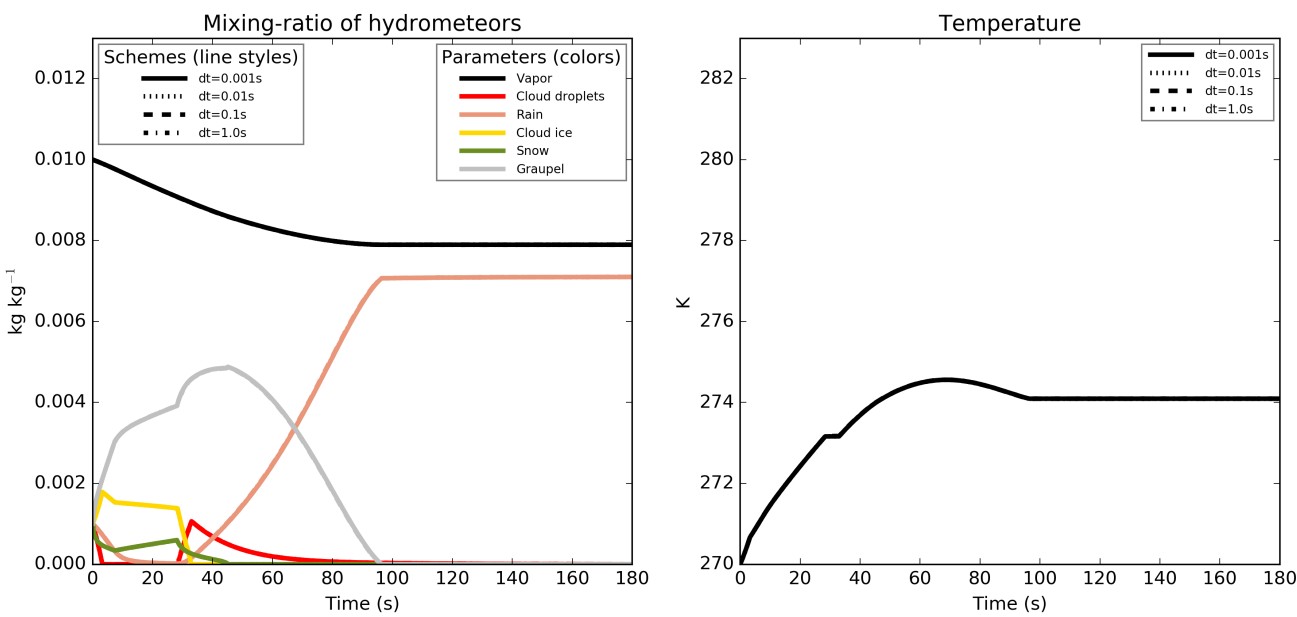

**Figure 9.** Same as Fig. 7 but using the new version of the microphysical scheme (all four curves are superimposed).

the ICE scheme. In order to remain stable, the Eulerian scheme uses a time-splitting technique with an upstream differencing

scheme. The internal time step[1] used for this computation is determined from the Courant-Friedrichs-Lewy (CFL) stability criterion based on a maximum fall velocity of $40 \, \mathrm{m \, s^{-1}}$ if hail is allowed or $10 \, \mathrm{m \, s^{-1}}$ otherwise (then, the same internal time step is used for all the species). These schemes take into account a terminal fall speed directly linked to the mean hydrometeor content (the more the content, the more rapid the fall). This feature makes it difficult, for these schemes, to accurately resolve the sedimentation process.

A vertical profile was initialized with a rain mixing-ratio of $0.1 \, \mathrm{g \, kg^{-1}}$ in one cell at $1400 \, \mathrm{m}$ above ground level; grid levels were $10 \, \mathrm{m}$ thick. In order to build a reference solution independent from the time step, a box-Lagrangian scheme (based on Kato, 1995) is used with the one-step time-advance method (see Sect. 2.2 for a description of the time-advance method). Because the model schemes use a particle size distribution, the first reference simulation we can build is through the dividing of the total content into bins and the application of the sedimentation on each bin (as for Milbrandt and McTaggart-Cowan, 2010). The reference time evolution is then shown in the upper panel of Fig. 10 (this reference is computed using only 25 bins to allow the reader to identify the trajectories of each bin; when more bins are used the time evolution gets smoother). The biggest drops reach the ground after around $200 \, \mathrm{s}$ whereas the smallest ones have yet to reach the ground after $1500 \, \mathrm{s}$. After $400 \, \mathrm{s}$ of simulation (dashed vertical line on the plot), one third of the rain is expected to be on the ground and the remaining part spread in the column. The two schemes in use in the model (BSB2010 and the Eulerian scheme) however compute the mass-weighted bulk terminal fall velocity and apply this velocity to the entire content. With this hypothesis, the expected time evolution is provided by the lower panel of Fig. 10. In this case, all the rain content is expected to follow the same trajectory and to be near the ground after $400 \, \mathrm{s}$ of simulation. The first reference simulation has a more realistic behavior and is able to reproduce the size-sorting effect but this result cannot be reached by the one-moment schemes used in this study. The bulk simulation is taken as the reference simulation with which to compare the model schemes over $400 \, \mathrm{s}$ long simulations. According to the initial mixing-ratio, to the parameters used in the sedimentation scheme and to the hypothesis of a mass-weighted bulk terminal fall velocity, the rain is expected to fall with a $3.3 \, \mathrm{m \, s^{-1}}$ speed, leading to a fall of $1320 \, \mathrm{m}$ during a $400 \, \mathrm{s}$ period. It should be noted that with a grid spacing of $10 \, \mathrm{m}$, the unit value for the CFL number is reached with a time step of around $3 \, \mathrm{s}$.

The top panel of Fig. 11 shows the resulting profiles for different time steps (600 simulations are performed using time steps between $0.1 \, \mathrm{s}$ and $60 \, \mathrm{s}$ with an increment of $0.1 \, \mathrm{s}$) using the BSB2010 scheme (this is not a time evolution, all profiles are the result of a $400 \, \mathrm{s}$ long integration). The time-step dependency is evident and it must be noted that for time steps longer than $30 \, \mathrm{s}$ a part of the water has reached the ground. The longer the time step is, the greater this part is. For the $60 \, \mathrm{s}$ time step, 11% of the total water has reached the ground. The BSB2010 scheme behaves differently regarding the CFL number. For CFL numbers larger than one, the diffusion on the vertical is more intense than for CFL numbers smaller than one. And the result obtained for a number of one is completely different from the results obtained with other values. The artifacts seen in the figure illustrate the shortcoming of the scheme but can not be as large in a true model simulation. In a real case with advection, irregular grid spacing, microphysical sources and sinks, the CFL value is not constant and each column of the model is a mixture of these different behaviors.

---

[1]the term "internal time step" is reserved, in this paper, for the time step used internally in the scheme to perform the time splitting. It is different from the (external) time step used for the scheme integration

For the Eulerian scheme (lower panel of Fig. 11), the time-splitting technique used with a very small internal time step of 0.25 s (value obtained considering a maximum fall speed of $40 \text{ ms}^{-1}$) leads to perform the same iterations whatever the time step of the simulation greater than this value. There is therefore no time-step dependency in the result. The very small internal time step induces however, as for the BSB2010 scheme with small values of the CFL number, an upward diffusion and a reduced fall speed. The Eulerian scheme, in addition, has an increased numerical cost (proportional to the number of iterations) due to the very small internal time step used.

Neither of these two schemes is correct; both diffuse the precipitation and have too low a fall speed. Additionally, the BSB2010 scheme exhibits a time-step dependency whereas the Eulerian scheme is costly. A compromise can be found by optimizing the Eulerian scheme in order to obtain a scheme without time-step dependency and with a reasonable cost. In the optimized version, at each iteration in the time splitting, the time of integration is computed from the maximum CFL number on each column and for each species instead of using a constant (in space and time) value.

The result is shown in Fig. 12 for different values of maximum CFL number allowed. With a maximum CFL number of one (top panel), the time-step dependency is quite large. With a maximum CFL number value of 0.1 (panel in the middle), the resulting figure is almost identical to the Fig. 11. The lower panel is for a larger maximum CFL number (0.8). For small time steps inducing a CFL number inferior to this maximum value, a time-step dependency can be seen but for large time-step values, the computation leads to the same results whatever the time step is. In contrast with the original version, this last case is less diffusive and is able to reproduce a peak value (in the bottom of the precipitation envelope). The resulting fall speed is still too small (after the 400 s integration duration, all the water content should be near the ground according to the hypothesis) but slightly better than those produced by the other versions of the Eulerian scheme.

In order to test further the impact of the algorithm on the sedimentation results, the box-Lagrangian scheme (used previously to build the reference results) is used in a simulation mode (using the step-by-step time-advance method). The top panel of Fig. 13 shows the resulting profiles after the 400 s long simulation using the bulk approach. The result is noisy, time-step dependent and there is still no rain on the ground. The noise could certainly be reduced by imposing a speed continuity between the different layers of the model (following the idea of Henry Juang and Hong (2010) for example). The box-Lagrangian scheme can also be used in an hybrid mode (as used in Morrison, 2012). At each time step, the content is divided into bins, each bin falls using the box-Lagrangian scheme and, at the end of the time step, the total content at each model layer is computed and used by the following iteration. The simulations are done using 500 bins (lower panel of Fig. 13). This scheme allows the bigger drops to fall more quickly. While this approach reduces the noise, some time-step dependency remains and the scheme is still not able to make the precipitation fall quickly enough to reproduce the reference simulation (this scheme also does not fulfill the requirement of a mass-weighted bulk terminal fall velocity used to build the reference). It should be noted nevertheless that this result is quite similar to the result obtained with the new version of the Eulerian scheme (lower panel of Fig. 12).

On one hand, the algorithm choice may well be important because the selection modifies the effective fall speed and the relative position of precipitation on ground with respect to the cloud which generated it, through the horizontal advection. On the other hand, none of the schemes tested in this study are able to reproduce the reference simulation due to the diffusion appearing during the fall which leads in turn to a reduced speed. Additionally, with the mixing induced by the dynamic and

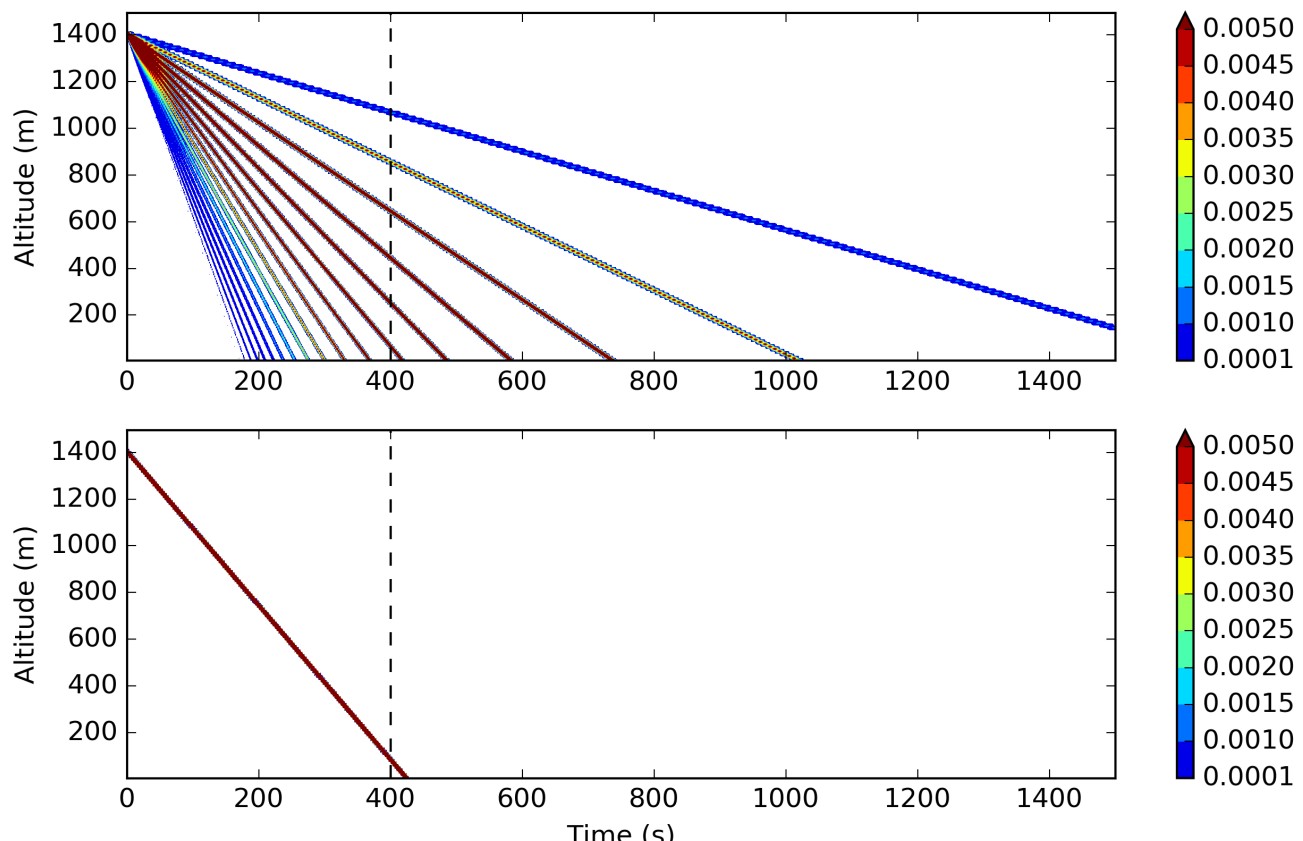

**Figure 10.** Time evolution of the vertical profile of rain mixing-ratio (the color scale represents the mixing-ratio in $\mathrm{g\,kg^{-1}}$) for a bin box-Lagrangian scheme (upper panel) and a bulk box-Lagrangian scheme (lower panel).

the turbulence, and with the interaction with the other microphysical processes, this choice has little impact on the resulting simulation of a real 3D case (not shown). It is believed that no scheme could perform drastically better with the one-moment hypothesis.

As from the conclusions of Milbrandt and McTaggart-Cowan (2010), one can expect better behavior from two-moment
5   schemes. This study could be extended in the future to the sedimentation used by the two-moment LIMA scheme.

This section illustrated how PPPY can be used to compare and exhibit the main behaviors of different 1D sedimentation schemes written in Fortran and Python using different time-advance methods.

## 4   Conclusions

In this paper, a new software designed to allow the comparison of Physical Parametrizations with PYthon (PPPY) independently
10   of all other model components was described technically and functionally. Its ability to use Fortran-compiled library from

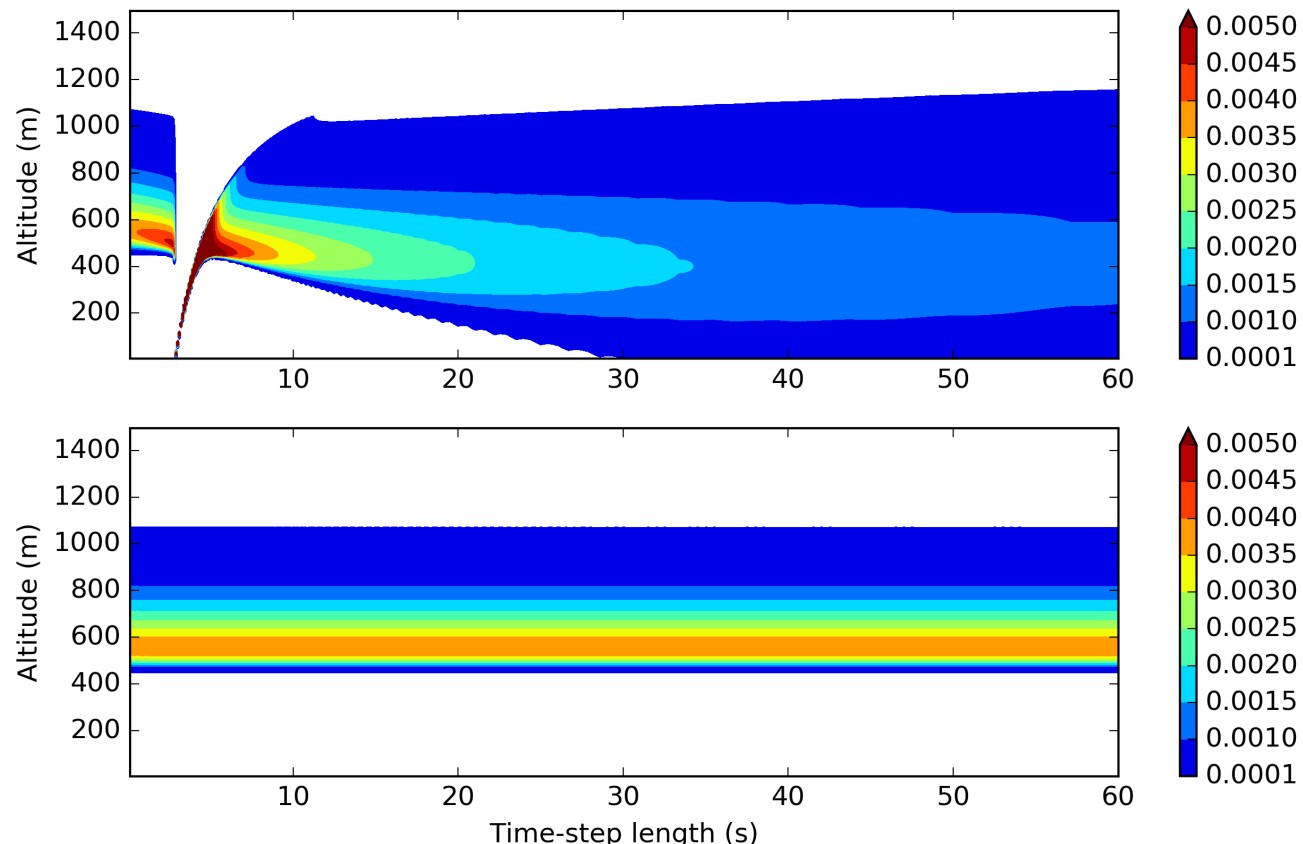

**Figure 11.** Vertical profiles of the rain mixing-ratio (the color scale represents the mixing-ratio in $\text{g kg}^{-1}$) after a 400 s long integration for different time steps (600 simulations are performed using time steps between 0.1 s and 60 s with an increment of 0.1 s, abscissa) for the BSB2010 scheme (upper panel) and the Eulerian scheme as available in the operational source code (lower panel).

different models, as well as Python based parametrizations, has been shown and used through two examples. It has been successfully used (in a 0D mode) to identify the sources of the time-step dependency which were present in the microphysical scheme in use in the AROME and Meso-NH models. Solutions have been proposed to correct the scheme and have been tested with the software. The sedimentation schemes have then been plugged and compared (in a 1D mode) to a reference box-5  Lagrangian scheme. These two examples have shown that it would be beneficial to use this kind of tool systematically when developing a parametrization in order to perform simple tests providing a first validation step (mass conservation, time-step dependency, absence of oscillations), before undergoing more complex validation stages (1D model, full simulations).

In addition to the ICE scheme, the LIMA scheme and some of the WRF microphysical schemes have been plugged. It could now be used to compare microphysical schemes originally hosted by different models (AROME, Meso-NH and WRF). In order 10  to compare these, it will be necessary to work on the initialization of the different schemes to ensure that the results can in fact be compared. In addition, it will be necessary to overcome the time-step dependency issue in order to allow the comparison of

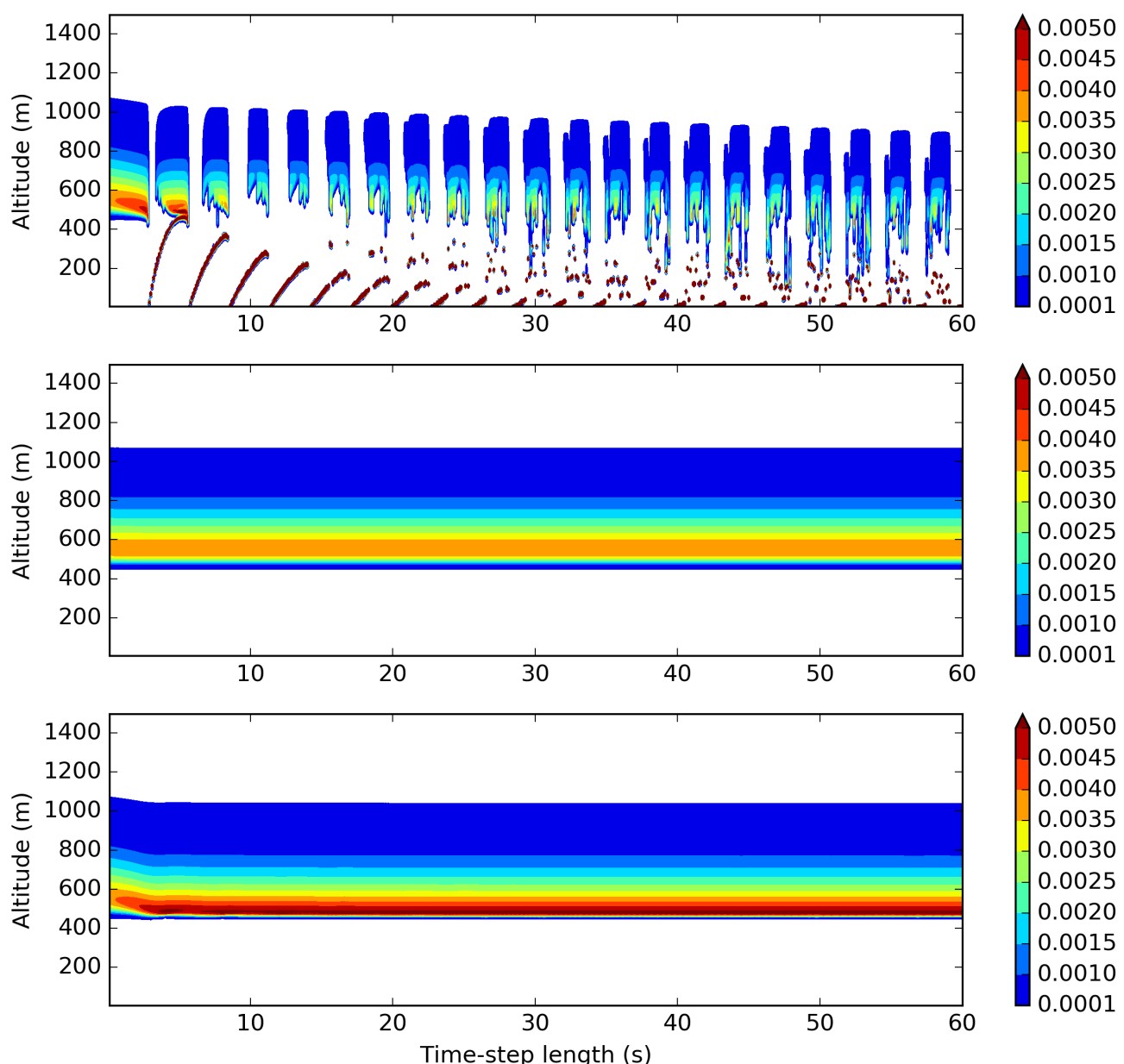

**Figure 12.** Same as Fig. 11 but for the modified version of the Eulerian scheme using different maximum values for the CFL number (1.0, 0.1 and 0.8 from top to bottom). The color scale represents the mixing-ratio in $\mathrm{g\,kg^{-1}}$.

the different schemes. One way to do this could be to make a compromise by selecting a time step that is not too large to be able to mask the time-step dependency and, at the same time, not too small to limit the computation cost.

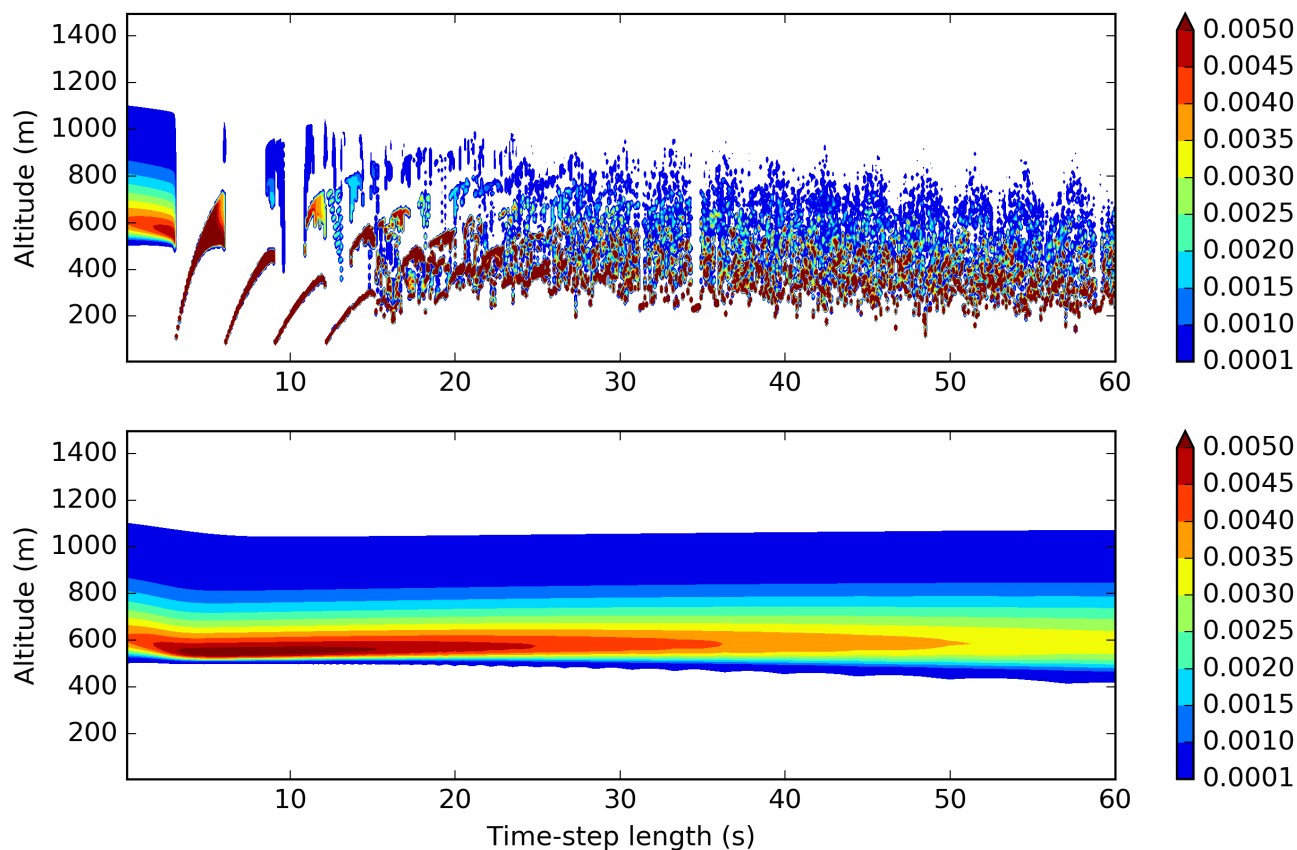

**Figure 13.** Same as Fig. 11 but for the box-Lagrangian scheme using the bulk approach (upper panel) and an hybrid approach (lower panel). The color scale represents the mixing-ratio in $\mathrm{g\,kg^{-1}}$.

PPPY is not limited to microphysical schemes and, in the future, it could also be used to compare other parametrizations such as mass flux or turbulence schemes.

*Code availability.* PPPY is freely available under CeCILL-C license agreement (a French equivalent to the L-GPL license; http://www.cecill.info/licences/C_V1-en.txt). PPPY v1.1 can be downloaded at https://doi.org/10.5281/zenodo.3490380.

## Appendix A: Simple example

Several examples of PPPY usage are provided with the software. Among them, a special example is intended to show how the different Python objects interact with each other and with the Fortran code; this is the test example which can be found in the `examples/test` directory of PPPY. This example is used in this appendix to illustrate the different steps needed to perform a parametrization comparison.

### A1 Compilation

Let's assume that the following code is put inside a file named `param.F90` and represents a model parametrization that we want to use with PPPY:

```
SUBROUTINE PARAM1(X, Y)
IMPLICIT NONE
REAL(KIND=8), INTENT(IN), DIMENSION(:, :) :: X
REAL(KIND=8), INTENT(OUT), DIMENSION(:, :) :: Y
Y=X+1
END SUBROUTINE PARAM1
```

It is suggested that the compilation procedure of the model be employed. One must therefore ensure that the normal compilation of the model builds a position-independent code, suitable for dynamic linking (`-fPIC` option). If not, the Makefile or the compilation script of the model must be updated to include such an option.

To use ctypesForFortran, a wrapper must be written to hide certain characteristics. The exposed dummy arguments:

- must not be of assumed shape or assumed rank (including string length) type;

- must not be optional;

- of Boolean type must be one-byte long.

In the test example, a wrapper (written in a file named `param_py.F90`) is needed to hide the assumed shape characteristic:

```
SUBROUTINE PARAM1_PY(X, Y, I1, I2)
IMPLICIT NONE
INTERFACE
    SUBROUTINE PARAM1(X, Y)
        REAL(KIND=8), DIMENSION(:,:), INTENT(IN) :: X
        REAL(KIND=8), DIMENSION(:,:), INTENT(OUT) :: Y
    END SUBROUTINE PARAM1
END INTERFACE
```

```
      INTEGER(KIND=8), INTENT(IN) :: I1, I2
      REAL(KIND=8), INTENT(IN), DIMENSION(I1, I2) :: X
      REAL(KIND=8), INTENT(OUT), DIMENSION(I1, I2) :: Y
      CALL PARAM1(X, Y)
5   END SUBROUTINE PARAM1_PY
```

When it is possible, it is suggested that this wrapper be included in the source code of the model in order to benefit from the Makefile or compilation script. If this proves impossible, the wrapper must be compiled outside the model environment but one must use the same compilation options for the wrapper as those used for the model.

Often, a parametrization must be initialized by calling a specific subroutine (in particular to set up constant values). In this example, this step is achieved by calling the following subroutine (which is also included in the `param_py.F90` file and does not need to be wrapped):

```
      SUBROUTINE INIT(ICONF)
      IMPLICIT NONE
      INTEGER(KIND=8), INTENT(IN) :: ICONF
15  END SUBROUTINE INIT
```

In addition, the example includes a `PARAM2` subroutine (copy of `PARAM1` except that $Y$ is $X + .9$) and an associated `PARAM2_PY` subroutine.

It is then supposed that a normal model compilation produces the compiled version of all these subroutines. In the example, the compilation is obtained (using gfortran) by the following command lines: `gfortran -c -fPIC param.F90` and `gfortran -c -fPIC param_py.F90`.

The last step of the compilation process is to build a shared library with `PARAM1_PY`, `PARAM2_PY` and `INIT` as entry points. This can be done through adding a compilation target in the model Makefile or compilation script, or by performing a manual build. In the example, the command line `gfortran -shared -g -o param.so param_py.o param.o` produces the `param.so` file.

## A2   The parametrization from the Python point of view

To use the previously build shared library from Python using the ctypesForFortran module, the following code is needed:

```
      import ctypesForFortran
      IN = ctypesForFortran.IN
      OUT = ctypesForFortran.OUT
30  INOUT = ctypesForFortran.INOUT

      ctypesFF, handle = ctypesForFortran.ctypesForFortranFactory('./param.so')
```

```
    @ctypesFF()
    def init(ICONF): #Name of the function must be the name of the actual fortran function
        "init function"
return ([ICONF],
               [(numpy.int64, None, IN), #INTEGER, INTENT (IN)
               ],
               None)

@ctypesFF()
    def param1_py(x):
        "Function that actually call the parameterisation"
        return ([x, x.shape[0], x.shape[1]],
               [(numpy.float64, x.shape, IN),
(numpy.float64, x.shape, OUT),
                (numpy.int64, None, IN),
                (numpy.int64, None, IN)
               ], None)
```

In this example, one of the Fortran subroutines is named `INIT` (case insensitive). By default, it is supposed that the compiled
object's name is the Fortran subroutine name (lowercase) with a trailing underscore (`init_` in this example)). If this is not the
case (because of a different compiler behaviour or a Fortran module use), a different prefix and/or suffix can be set in argument
of the `ctypesFF` function of the example to obtain a decorator able to find and call the Fortran code.

The Python function must return three elements:

  – the list of the values expected in input by the Fortran subroutine;

– the list of the dummy arguments of the Fortran subroutine;

  – the type of the returned value for a Fortran function (None for a subroutine).

Each dummy argument is described by a tuple: type of the argument expressed as a numpy type, shape (or None for a scalar) of
the argument and input/output status. More examples are available inside the ctypesForFortran module source code. It should
be noted that this part can be replaced by the use of other interfacing method such as f2py.
To use the parametrization with PPPY, a Python class must be written (see the `pppy_param1.py` file of the example for
a complete implementation) by inheritance: `class pppy_param1(pppy.PPPY):`. The class contains a __call__ method
which calls the different methods in this order: setup, build_init_state, execute (which is called in a loop) and lastly finalize.

The methods that can or must be implemented to represent a parametrization are described below (only the execute method is mandatory):

- __init__: This method can be implemented to deal with the possible parametrization options.

- setup: This method does the initialization part that cannot be done earlier (in the __init__ method) or needs to be done again before each of the simulations. In the various provided examples, this is the place where the shared library is opened, where signatures of Fortran routines are defined and where the initialization of the Fortran modules are performed.

- finalize: This method can be useful for cleaning the memory or disk after running a simulation.

- build_init_state: When performing a comparison, each of the parametrizations is called with the same initial state. This method is the place at which to add state or diagnostic variables specific to the parametrization.

- execute: This method calls the parametrization to perform a time advance.

More details about these methods are given in the PPPY documentation.

## A3 Comparison

The comparison is performed by the `comp_test.py` file.

Firstly, the parametrizations must be defined by choosing the time step (dt argument), the time-advance method (method), the names (the name argument is used for the plot labels and the tag one for building file names) and the possible options (solib and iconf here). For the example described here, this is done by the following lines:

```
param_1 = pppy_param1(dt=60., #time step to use with this parametrization
                      method='step-by-step', #like a true simulation
                      name="Param #1", #name to use for plots
                      tag="param_1", #tag to use for file names
                      solib=solib, #1st pppy_param1 option: shared lib file name
                      iconf=iconf) #2nd pppy_param1 option: configuration
```

Several parametrizations can be defined by changing the time step, the time-advance method or the options, using the same source code or not.

The comparison is defined and the simulations are performed by:

```
comp = PPPYComp(schemes=[param_1, param_2], #List of parametrizations to compare
                output_dir=output_dir, #directory to store the results (hdf5 files)
                duration=180., #duration of simulation
                init_state=dict(x=numpy.array([0.])), #initial state
```

```
                        name="First test", #name to use in plots
                        tag="firstTest") #tag to use for file names
    comp.run()
```

A simple plot is obtained with:

```
5   plot = ('evol', #time evolution
            dict(var_names=['x'])) #of variable x
    fig, plots = comp.plot_multi((1, 1), #only one plot
                                  [plot])

    plt.show()
```

## 10   Appendix B:  Main modifications needed to suppress the time-step dependency in the ICE scheme

The most important modifications that were needed to suppress the time-step dependency in the ICE scheme are listed above:

- heat budgets must be computed when the feedback on temperature can stop the process. For example, when the temperature exceeds 0 °C, a species cannot melt more than the quantity that would imply a temperature below 0 °C. The lack of heat budgets explain a large part of the time-step dependency observed with the simulations using the small time steps
(shown in Fig. 7);

- in the original version of ICE, the snow content rimed by cloud droplets (to produce graupel) was computed as an adjustment: the process provided the mass of snow to convert into graupel and this mass was then divided by the time step. The mass of transformed snow did not take into account the quantity of cloud water involved. These two characteristics were at the origin of a time-step dependency. The process was modified using the Murakami (1990) approach based on
the comparison between the effective cloud droplets collection and the mass of water needed to transform low density snow into high density graupel;

- the graupel growths mainly by collecting other species. When this collection implies liquid species (rain and/or cloud), there are two possibilities (called growth mode) depending on a heat balance: the graupel is able to freeze the entire collected liquid (dry mode) or a thin liquid film appears at the graupel surface (wet mode). In the original version of the
ICE scheme there was confusion between the maximum content of liquid water than can be frozen (which must be used in the heat balance) and the content of liquid and ice water that can be collected in wet growth mode (which must be used to compute the graupel tendency). The correction made the mode choice more continuous. And because, ultimately, the graupel growth mode has an impact on the collection efficiency of icy species (snow and cloud ice) with the graupel, this choice can lead to significant differences in the collection rates. Hence, the scheme including the correction is less
time-step dependent;

- the water shedding (cloud droplets becoming rain drops when collected and not frozen by the graupel) was activated only for temperature below 0 °C whereas the process must be active as long as the graupel exists (when the graupel has melted into rain, rain drops actually collect cloud droplets; the process must be continuous during graupel melting);

- several modifications have been carried out on the processes involving the hail category as a prognostic field: the processes dealing with hail are now completely symmetric with those dealing with the graupel category (to ensure consistency even if this did not produce time-step dependency). A conversion fraction is computed from the heat balance used to choose the graupel growth mode. In the original version of the scheme, this fraction was applied on the total content of graupel; this induced a conversion tendency directly linked to the number of times the rate is applied (hence to the time step for a given simulation length). On the contrary, in the new version, the conversion fraction is applied on the wet growth rate, this way, no time-step dependency is produced. This was the main reason for the time-step dependency on the hail category.

The modifications listed above aim at suppressing the time-step dependency present inside each of the microphysical processes. These modifications were sufficient to suppress or, at least, limit the dependency for time steps up to 10 s (not shown). For greater time steps, each process must take into account that a given species can be consumed or produced, at the same time, by another process and that, therefore, this affects its efficiency. To address this issue, some kind of splitting was needed to reduce the effective time step used in the microphysical scheme allowing processes to interact more frequently.

The modified scheme allows two splitting methods: a classical time-splitting method that uses a fixed sub-time-step and a "mixing-ratio-splitting" method that computes, at each iteration, the sub-time-step to use in order not to have a single mixing ratio change exceeding a given threshold. The second method has the advantage of adapting the number of iterations to the intensity of the microphysical processes. When little happens, only one iteration is performed; on the contrary, when the exchanges between species are intense, several iterations are performed.

To produce Fig. 9 and Fig. 8, the threshold used in the mixing-ratio-splitting (0.01 $\mathrm{g\,kg^{-1}}$) was small and would induce a substantial additional cost on a real simulation. This value was chosen to illustrate the scheme behavior but a value of 0.05 $\mathrm{g\,kg^{-1}}$ seems to be more acceptable (empirical value obtained in comparing cost and time-step dependency in a 2D simulation) for operations. The iterations needed for the 0.05 $\mathrm{g\,kg^{-1}}$ threshold induce a cost increase of about 5% (in comparison with a simulation performed without splitting), and for the 0.01 $\mathrm{g\,kg^{-1}}$ threshold an additional 20% can be expected (this last figure is an estimation because no sensitivity test, regarding this threshold, has been performed on the whole domain used for operations).

*Competing interests.* The author declares that he has no conflict of interest.

*Acknowledgements.* I wish to thank B. Vié for having plugged in the LIMA scheme into PPPY, and the trainee team which worked on the WRF plugging in: A. Riandet, L. Richecoeur and M.-L. Roussel. I also thank C. Lac and B. Vié for their helpful comments on the manuscript.

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
