# Peer review of "Development of "Physical Parametrizations with PYthon" (PPPY, version 1.1), and its usage to reduce the time-step dependency in a microphysical scheme"

_Geoscientific Model Development, 2019_

## Referee Comment (RC1) · Anonymous Referee #1 · 23 Jun 2019

The paper describes a workflow and a set of Python scripts for debugging isolated units of large-scale (i.e., NWP/GCM) atmospheric models that are predominantly written in Fortran. The presented workflow focuses on subgrid or column-wise parameterisations of physical processes. The key highlighted feature of the presented tool, named PPPY, is the ability to interact with units of unmodified compiled Fortran code. As hinted in the title, the work was motivated by the goal of exploring time-step dependency of the small-scale process representations. The paper and the supplemented source code depicts the ability to employ the tool with multiple microphysical schemes

and multiple simulation frameworks including AROME/Meso-NH and WRF. There are two case-study analyses presented focusing on the ICE cloud microphysics parameterisation and on the BSB2010 hydrometeor sedimentation scheme. Discussed Python scripts are available for download and released under an open-source license.

My assessment of the manuscript is generally negative. Although the key deficiencies lie in the content, it is hard not to mention first that the manuscript seems to have been submitted prematurely. The abstract and first paragraph end with ellipses (!), there are abundant grammatical errors and typos, oftentimes the language used is casual (e.g., "*I will just list rapidly the most important ones*") and numerous statements are strikingly vague (e.g., "*the tool can be used in 0D and 1D mode, with schemes coming from different models and with different time-advance methods to produce different kind of plots*"; "*it led to problems maybe specific to our environment or source code*").

The content of the paper encompasses atmospheric modelling techniques and technicalities of Python-Fortran interfacing. None of these two subjects are covered in sufficient detail in my opinion (perhaps focusing on just one of those would be a path forward)? Noteworthy, already the title of the paper implies description of improvements to a particular microphysical scheme. These improvements, described in section 3.1, are presented as purely textual description with vague statements, e.g. „*the graupel growth mode choice was updated. It is now more continuous and, hence, less time-step dependent*". Such approach does not match GMD's standards aimed at clarity with respect to model formulation and versioning. The author does explicitly state that the "*purpose of the paper is not to give an extended review of all the modifications*", yet in my opinion the way the model development is documented in the paper goes against GMD policies.

The Python-Fortran interfacing subject, covered in section 2, is presented with similar level of vagueness. The key components of the presented software included in the 11k LOC ctypesForFortran.py file are not discussed at all. Overall, I expect that independent use of the presented PPPY package, would not be easier than obtaining analogous functionality "from scratch" using a general-purpose Python package providing abstractions for interfacing compiled code (e.g., CFFI which has numerous documented examples depicting its usage with Fortran code and NumPy arrays).

Although, in principle, I would be reluctant to call something "too basic", reading the manuscript I felt puzzled with regard to the intended audience of the paper. I feel confident that GMD readers do not require repeated verbose explanations on what numerical diffusion is and why it vanishes for integer Courant numbers. The same concerns such statements as: „*Python was chosen because it allows to make plots ...*", "*the computational time can be large when very small time steps are used*" or "*One process must take into account that a given specie can be consumed or produced, in the same time, by another process*".

Below, I am listing some more specific comments that perhaps can be helpful for the author, and that support my opinion outlined above:

- avoid frequent use of the word "tool" (over 40 occurrences including all but one sentences of the abstract)

- avoid ellipses

- do not use programming notation such as "1.E-5" in the text

- time step vs. time-step, etc - please be consistent

- ensure the use of the words "statistical" and "physical" is justified for all its occurrences

- please do not call something "classical" without reason

- following phrases have certainly better alternatives: "home made", "some behaviors of a scheme", "object made from a class", "intensity of the 0D simulations",

"more the content is important, more the fall is rapid", "weak content", "leads to do approximately the same computation", "content is artificially put higher"

- capitalise Python

- use vector graphics for figures

- ensure consistency in bibliographic entries: abbreviated (with dots or without) and non-abbreviated journal names

---

## Referee Comment (RC2) · Andrew Barrett (Referee) · 5 Jul 2019

Review of `Development of "Physical Parametrizations with Python" (PPPY, version 1.1), and its usage to reduce the time-step dependency in the ICE microphysical scheme' by Sebastien Riette

General comments:

This paper gives an overview of a new python tool to evaluate parameterizations used in numerical models, and in particular to address the issue of time step dependency of results in the Meso-NH model. The tool is first used to evaluate the time step dependency in the cloud microphysical parameterizations of Meso-NH, and additionally several other microphyics schemes used in the WRF model. The causes of the time step dependency are highlighted and a revised version of the model is later evaluated. This is achieved through using the tool in 0D mode. Later, the tool is extended to a 1D simulation, where two sedimentation schemes are evaluated against a more sophisticated reference.

The paper addresses an important area – of changing behavior of numerical parameterizations when used with different timesteps, or with under different forcings. The area is becoming increasingly important within the area of Physics-Dynamics coupling. Therefore such a tool could provide great benefit to the community, especially because it enables the use of existing model code (in fortran), despite the tool being coded in python.

Overall, the paper is well written and clear. However, there are areas where more discussion would be very helpful for the reader (and potential tool user) – in particular in the introduction of the tool and description of how to apply it for existing model code. Additionally, I find the lengthy discussion of the errors and biases in sedimentation schemes in the second half of the paper to be too much. There are numerous papers that already evaluate a diverse range of sedimentation schemes and this paper needs not repeat some of that. It would be sufficient to demonstrate the applicability of the tool in such a situation. I strongly recommend the publication of this paper, after minor revisions.

Specific comments:

1. As a potential future user of this tool, I would appreciate a greater and clearer introduction of how the tool works. The language is still quite technical in places and certain terms (e.g. "objects", "libraries" and "decorators") are likely not well understood by future readers. While I acknowledge that there is extra information in the documentation of the tool, I suggest adding some more description to the paper. In particular I would like to see some more advice regarding the interfacing of the fortran code to the python tool. The current description sounds rather ad-hoc and I'm not clear how I would replicate this method.

2. There is no description in the paper as to how this tool was used to identify and fix the causes of the time step dependence. The findings themselves are listed on page 9. Some additional description of how this tool enabled these model parameterization errors to be found and fixed is needed. This is the main benefit of this new tool, so it would be useful to see how it should be used.

3. This paper is not the place to discuss in detail the merits of different (and in this case rather simple) sedimentation schemes, therefore I suggest shortening this section to allow for the above expansions.

4. The figures in this paper (which, I think, are produced by the PPPY tool itself) are missing units on all axes. Additionally, Figure 3 does not make clear which parameterization scheme is shown in which panel. There two failings need to be fixed. Ideally, not just in the paper, but also in the code of the tool itself (for future users benefit).

5. There seems to be inconsistent model forcing used for the different schemes (or inconsistent physics regarding condensation in the different schemes). Comparing figures 1, 2 and 3 – the water vapor content (black line) for the shortest time step (1s) converges on a value of around 8 g/kg after 180 seconds, whereas for the other schemes in figure 2 and 3, the water vapor converges on 6 g/kg. What is the reason for this difference? Can it be corrected? I understand that the microphysics schemes will give different results for the hydrometeor concentrations, but the simple balance between temperature, water vapor and condensed water should be more or less the same for all schemes. At the moment it is difficult to compare results between the different figures/microphysics schemes.

Minor comments:
6. page 5, figure 3. It is impossible to tell from figure 3 and the caption, which of the subpanels relate to which microphysics scheme. This should at minimum be added to the caption, and preferable to the figure panels too.
The label "Schemes (line styles)" in the top left of each figure could easily be replaced by (e.g.) the scheme name/abbreviation for each panel.
The comparison of different microphysics schemes in figure 3 is initiated with a rather unrealistic setup (approximate relative humidity is 165%). In full (3D/4D) model simulations, such supersaturation would never occur, and the microphysics schemes should not be expected to treat such situations fully realistically. Nevertheless, I find the differences between the schemes very interesting – I would be particularly interested to see how these same schemes performed in more realistic setups (e.g. with significantly reduced supersaturation at t=0 and/or when a constant cooling rate is applied)

7. page 8, bullet points lines 9-28. These are all very interesting findings, but how did the PPPY tool help you to discover these factors as being important. It would be good to show the benefits of the tool you developed in achieving these findings. Was it simply a trial-and-error process, or is there some aspect of PPPY that enables these errors to be determined more quickly?

8. page 8, line 27-29 please clarify what you mean by "the conversion rate of graupel into hail is now computed from the wet growth rate of graupel and not from the total content of graupel". How large is this difference and why does it make such a difference?

9. Figures 11, 12 & 13. please make clear that the x-axis is timestep length (dt) – it could also be interpreted as timestep number (i.e. as a time-height plot)

Technical/language corrections:
- page 2, line 9-10. please provide mode details about what differences were seen in the Meso-NH model when the time step was changed?
- page 3, line 9. what is a "tool package". Where can the reader find it?
- page 3, line 13. "makes possible" → "makes it possible"
- page 4, line 1 correct to "consists of a python package written to ..."
- page 4, line 5 delete "which is the required"
- page 6, line 3 "needed to use the parameterization" → "needed by the parameterization"
- page 6, line 6 "that not exist" → "that do not exist"
-page 7, line 4 "to plug other" → "to plug in other"
- page 7, line 5 "one have to define" → "one has to define"
- page 8, line 2 "dependency on the simulation" → "dependency **of** the simulation"
- page 8, line 11 please quantify what you mean by "small time steps"

- page 8, line 18 it is not clear to me what you mean by "graupel growth mode", please give more details
- page 8, line 23 "has melt into rain" → "has melted into rain"
- page 8, line 25 "insure" → "ensure"
-page 9, line 10 & 11. please use standard scientific notation (e.g. $1.0 \times 10^{-5}$)
- page 9, line 11. please quantify the "substantial additional cost"
- page 9, line 16 "unique" → "single"
- repeated grammatical error (e.g. page 12, line 9; page 12, line 31; page 13, line 1, page 13, line 7): "Longer is the time step, more this part is important" → "The longer the time step is, the more important this part is"
- page 12, line 31. What is the "it" in "it reaches around 11%"?
- page 13, line 5 "mean content is weaker" → "mean content is less"
- page 13, line 20 "by consequences" → "as a consequence"
- page 13, line 24 "this induces" → "this means"
- page 13, line 25 "larger to one" → "larger than one"
- page 14, line 3 "None of both schemes" → "Neither of these two schemes"
- page 14, line 9 "weaker" → "less"
- page 14, line 28 "whatever is the time step" → "whatever the time step is"
- page 14, line 32 "hypothesis done" → "hypothesis"
-page 15, line 2 "certainly reduce" → "certainly reduced"
- page 15, line 10 "This scheme allows to make fall the bigger drops quicker" → "This scheme allows the bigger drops to fall more quickly"

---

## Author Comment (AC1) · 18 Oct 2019

The comment was uploaded in the form of a supplement:
https://www.geosci-model-dev-discuss.net/gmd-2019-111/gmd-2019-111-AC1-supplement.zip
* * *

---

## Author Comment (AC2) · 18 Oct 2019

The comment was uploaded in the form of a supplement: https://www.geosci-model-dev-discuss.net/gmd-2019-111/gmd-2019-111-AC2-supplement.zip

---

## Author Response (AR1)

**Reply to referee #1**

I thank anonymous referee #1 for his/her comments which have improved the manuscript. New manuscript text is italicized in the replies.

**Comment**: The abstract and first paragraph end with ellipses (!), there are abundant grammatical errors and typos, oftentimes the language used is casual (e.g., "I will just list rapidly the most important ones") and numerous statements are strikingly vague (e.g., "the tool can be used in 0D and 1D mode, with schemes coming from different models and with different time-advance methods to produce different kind of plots"; "it led to problems maybe specific to our environment or source code").

**Response**: The manuscript has been corrected by a native writer of English.

**Comment**: The content of the paper encompasses atmospheric modelling techniques and technicalities of Python-Fortran interfacing. None of these two subjects are covered in sufficient detail in my opinion (perhaps focusing on just one of those would be a path forward)? Noteworthy, already the title of the paper implies description of improvements to a particular microphysical scheme. These improvements, described in section 3.1, are presented as purely textual description with vague statements, e.g. „the graupel growth mode choice was updated. It is now more continuous and, hence, less time-step dependent". Such approach does not match GMD's standards aimed at clarity with respect to model formulation and versioning. The author does explicitly state that the "purpose of the paper is not to give an extended review of all the modifications", yet in my opinion the way the model development is documented in the paper goes against GMD policies.

**Response**: The goal of the paper is to present the 0D tool. The ICE microphysical scheme and the modifications that have been applied on it are only there to illustrate the PPPY behaviours. To make it more clear, I suppressed the name of the scheme from the title, the abstract is somewhat rewritten and the improvements to the ICE scheme have been moved to an appendix to suppress them from the manuscript body.

Moreover, some details have been added in the section 2.1.2, in addition to the ctypesForFortran details in section 2.1.1 (in response to your comment below), to improve the OD presentation.

**Comment**: The Python-Fortran interfacing subject, covered in section 2, is presented with similar level of vagueness. The key components of the presented software included in the 11k LOC ctypesForFortran.py file are not discussed at all. Overall, I expect that independent use of the presented PPPY package, would not be easier than obtaining analogous functionality "from scratch" using a general-purpose Python package providing abstractions for interfacing compiled code (e.g., CFFI which has numerous documented examples depicting its usage with Fortran code and NumPy arrays).

**Response**: I do not consider that ctypesForFortran is the key component of the software because PPPY users can use ctypesForFortran, f2py, directly ctypes or another tool such as CFFI. However, I included a short description of the main features ctypesForFortran includes:

> *The PPPY user is free to use whichever Python-Fortran interfacing method he chooses (among the two aforementioned or other ones). The ctypesForFortran way intends to help the interfacing of Fortran functions and subroutines on a Linux system. It handles memory allocations and array memory order. Internally ctypesForFortran uses the Python ctypes module (which normally handles the C shared libraries) to interact with the library without*

*adding a C or Fortran layer. It deals with Boolean, strings, integers and floats (32- and 64-bits) but does not support structures. The array and string arguments must be explicitly defined (no ``:``, ``..'' or ``*'' are allowed in the interfaces) and no argument can be optional. If this is not the case, a wrapper must be written in Fortran meeting these requirements and calling for the original subroutine.*

In addition, to be more concrete, I included an annexe to give an example of PPPY usage.

When we encountered problems with f2py, we found easier to bypass these problems by writing the ctypesForFortran module that we can control. Maybe it exists a universal interfacing tool which is suitable in all circumstances but a rapid test shows that CFFI also brings problems concerning boolean scalars with some compilers (the binary representation of a Boolean scalar with Fortran is different depending on the compiler, eg. intel vs gfortran). I expect that this problem can be solved with a Fortran 2003 compatible compiler using "bind(c)" but this was not an option when we wrote ctypesForFortran.

**Comment**: Although, in principle, I would be reluctant to call something "too basic", reading the manuscript I felt puzzled with regard to the intended audience of the paper. I feel confident that GMD readers do not require repeated verbose explanations on what numerical diffusion is and why it vanishes for integer Courant numbers. The same concerns such statements as: „Python was chosen because it allows to make plots ...", "the computational time can be large when very small time steps are used" or "One process must take into account that a given specie can be consumed or produced, in the same time, by another process".

**Response**:

- For the verbose explanations on numerical diffusion: Sect. 3.2 have been rewritten

- For the python choice: I'm sure many readers know that Python can make plots but what is important is that a **single** language can produce a plot **and** interact with a compiled code. In the revised manuscript, two sentences have been merge to be more concise and to not appear to be too basic:
  *The tool consists of a Python package which drives the simulations and performs the comparison: initialization, the calling of the Fortran routines (using the original source code of the parametrization), the saving of the results (in HDF5 files using the h5py module) and the plotting of the results (through the matplotlib module).*

- The remark about the computational time that can be large despite of being in a 0D mode is suppressed.

- For the interaction between processes, this can appear to be too basic but it is important to mention it because 1) this interaction is not taken into account in a number of microphysical schemes (except by preventing negative values for the hydrometeors) and this induces an uncertainty on the results, and 2) this is the reason why the splitting was introduced in the ICE scheme. I slightly reordered the sentences to exhibit more the relation between the interactions and the splitting:
  *The modifications listed above aim at suppressing the time-step dependency present inside each of the microphysical processes. These modifications were sufficient to suppress or, at least, limit the dependency until time steps around 10 s (not shown). For greater time steps, each process must take into account that a given species can be consumed or produced, at the same time, by another process and that, therefore, this affects its efficiency. To address this issue, some kind of splitting was needed to reduce the effective time step used in the microphysical scheme.*

**Comment**: Below, I am listing some more specific comments that perhaps can be helpful for the author, and that support my opinion outlined above:

- avoid frequent use of the word "tool" (over 40 occurrences including all but one sentences

of the abstract)

- avoid ellipses
- do not use programming notation such as "1.E-5" in the text
- time step vs. time-step, etc - please be consistent
- ensure the use of the words "statistical" and "physical" is justified for all its occurrences
- please do not call something "classical" without reason
- following phrases have certainly better alternatives: "home made", "some behaviors of a scheme", "object made from a class", "intensity of the 0D simulations", "more the content is important, more the fall is rapid", "weak content", "leads to do approximately the same computation", "content is artificially put higher"
- capitalise Python
- use vector graphics for figures
- ensure consistency in bibliographic entries: abbreviated (with dots or without) and non-abbreviated journal names

**Response**: I did my best to take into account your remarks. Some of them need a specific reply:

- 1.E-5 was an error, this is corrected in the text
- The manuscript has been corrected by an English native writer
- The graphics are outputs from PPPY, they are png files

**Reply to referee #2**

I thank Andrew Barrett for his comments which have improved the manuscript. New manuscript text is italicized in the replies.

**Comment 1**: As a potential future user of this tool, I would appreciate a greater and clearer introduction of how the tool works. The language is still quite technical in places and certain terms (e.g. "objects", "libraries" and "decorators") are likely not well understood by future readers. While I acknowledge that there is extra information in the documentation of the tool, I suggest adding some more description to the paper. In particular I would like to see some more advice regarding the interfacing of the fortran code to the python tool. The current description sounds rather ad-hoc and I'm not clear how I would replicate this method.

**Response**: I modified Sec. 2.1 to be more didactic:

> *Two kinds of objects exist: those which represent a parametrization, and those representing the comparison. A standard object (an abstract class) is provided in order to define a parametrization (the PPPY box in Fig. 4). This abstract class already contains everything needed to perform the time advance and the saving of results but must be complemented (by inheritance) to incorporate the actual call to the different parametrization codes (Param1 and Param2 boxes of the figure). Finally, each parametrization can be used with different configurations. To achieve this, different instances (Param1.1, Param2.1 and Param2.2 boxes) are created, one for each of the configurations (e.g. time-step length, options specific to the parametrization).*
>
> *For the comparison, the provided class (PPPYComp in the figure) can be used directly or can be complemented (by inheritance, UserComp box in the figure) to add new diagnostics (e.g. new plot kind, computation of a derived variable to plot). An instance of the class is created for each comparison to perform (Comp box). A comparison is defined by the list of the parametrizations to use, the simulation length and the initial state. This comparison instance drives the parametrization instances to carry out the simulations and to plot the result.*

Moreover, an appendix is added to describe with more details the test example which is provided with PPPY. I think it is a good entry point to understand how the tool works before going through the other examples which are more interesting but also more complicated.

**Comment 2**: There is no description in the paper as to how this tool was used to identify and fix the causes of the time step dependence. The findings themselves are listed on page 9. Some additional description of how this tool enabled these model parameterization errors to be found and fixed is needed. This is the main benefit of this new tool, so it would be useful to see how it should be used.

**Response**: Please see comment #7

**Comment 3**: This paper is not the place to discuss in detail the merits of different (and in this case rather simple) sedimentation schemes, therefore I suggest shortening this section to allow for the above expansions.

**Response**: I reduced the sedimentation section.

**Comment 4**: The figures in this paper (which, I think, are produced by the PPPY tool itself) are missing units on all axes. Additionally, Figure 3 does not make clear which parameterization

scheme is shown in which panel. There two failings need to be fixed. Ideally, not just in the paper, but also in the code of the tool itself (for future users benefit).

**Response**: The unit on the time and altitude axis were indeed missing, I added them. And, I moved the unit of the plotted variable from the title to the y-axis (the PPPY user can already define the title and the x/y labels). In addition, I changed the title of the different panels of Figure 3.

**Comment 5**: There seems to be inconsistent model forcing used for the different schemes (or inconsistent physics regarding condensation in the different schemes). Comparing figures 1, 2 and 3 – the water vapor content (black line) for the shortest time step (1s) converges on a value of around 8 g/kg after 180 seconds, whereas for the other schemes in figure 2 and 3, the water vapor converges on 6 g/kg. What is the reason for this difference? Can it be corrected? I understand that the microphysics schemes will give different results for the hydrometeor concentrations, but the simple balance between temperature, water vapor and condensed water should be more or less the same for all schemes. At the moment it is difficult to compare results between the different figures/ microphysics schemes.

**Response**: In the ICE scheme, condensation is apart from the other microphysical processes. I first checked the time-step dependency in the condensation part. If condensation is activated (with one call to the subroutine by time step), it tends to hide somewhat the time-step dependency by making the different simulations to converge towards an equilibrium point. In particular with the ICE scheme with the same setup as used in the manuscript, the adjustment would suppress the cloud ice, and thus the time-step dependency of all the processes involving this specie will not be seen. Several setups would then be necessary to explore all the microphysical processes. But, indeed, with the adjustment, the vapour content of ICE with the same setup converges towards 6g/kg.

[Figure]

*Illustration 1: Same as Fig. 1 of the manuscript but with the adjustment activated*

For the LIMA scheme, adjustment is active, it would be easy to deactivate it. But, for the WRF schemes, the condensation process is embedded inside the microphysical parametrizations.

For the current study, I must deactivate the adjustment in ICE to fully explore all the microphysical processes. Because the goal is not to compare the results between schemes but to compare, for each scheme, the simulations for different time steps, I think it is not very important if the setup and/or the active processes differ. In a next study I hope to be able to compare the different schemes; in this next study it will be necessary to pay attention to the setup and to the active processes.

I made several modifications in the text to exhibit more this difference between the ICE scheme and the others:

- *(excluding the saturation adjustment and the sedimentation)*
- *In the simulations performed with this scheme (Fig. 2), the setup is the same as for the ICE scheme but the saturation adjustment is active.*
- *The WRF simulations are performed using the saturation adjustment included inside each scheme.*

**Comment 6**: page 5, figure 3. It is impossible to tell from figure 3 and the caption, which of the subpanels relate to which microphysics scheme. This should at minimum be added to the caption, and preferable to the figure panels too. The label "Schemes (line styles)" in the top left of each figure could easily be replaced by (e.g.) the scheme name/abbreviation for each panel. The comparison of different microphysics schemes in figure 3 is initiated with a rather unrealistic setup (approximate relative humidity is 165%). In full (3D/4D) model simulations, such supersaturation would never occur, and the microphysics schemes should not be expected to treat such situations fully realistically. Nevertheless, I find the differences between the schemes very interesting – I would be particularly interested to see how these same schemes performed in more realistic setups (e.g. with significantly reduced supersaturation at t=0 and/or when a constant cooling rate is applied)

**Response**: See comment #4 for the labels.

The setup used in the paper was chosen to allow for a maximum of microphysical processes to be active during a single simulation even if it is not fully realistic. Changing the setup by reducing the vapour content at t=0 (to use 4g/kg instead of 10g/kg) reduces the number of existing species for the ICE scheme (and hence the number of active processes) but does not suppress the time-step dependency. I think the figures with this new setup are less illustrative, I prefer keeping the old ones in the manuscript but you will find the new ones below:

[Figure]

*Illustration 2: Same as Fig. 1 of the manuscript but with a drier setup*

[Figure]

*Illustration 3: Same as Fig. 2 of the manuscript but with a drier setup*

[Figure]

*Illustration 4: Same as Fig. 3 of the manuscript but with a drier setup*

I added a sentence in the manuscript to inform about the unrealistic setup:

> *The setup is not fully realistic (with an important supersaturation) but allows simulations to involve all the species and, hence, virtually all the microphysical processes. It was checked (not shown) that the time-step dependency still exists when using more realistic initial values.*

**Comment 7**: page 8, bullet points lines 9-28. These are all very interesting findings, but how did the PPPY tool help you to discover these factors as being important. It would be good to show the benefits of the tool you developed in achieving these findings. Was it simply a trial-and-error

process, or is there some aspect of PPPY that enables these errors to be determined more quickly?

**Response**: Unfortunately, I didn't find a better way than using a trial and error process (by enabling or not the different microphysical processes).

> *The simulations have been carried out several times activating and deactivating the different microphysical processes. To do this, the processes have been called individually by the PPPY software (when they are written in separate subroutines) or activated through switches or, at worst, (un-)commented in the source code. This trial-and-error process makes it possible to identify the processes that led to the oscillations and to the time-step dependency, and allowed the checking of each correction individually from the others.*

**Comment 8**: page 8, line 27-29 please clarify what you mean by "the conversion rate of graupel into hail is now computed from the wet growth rate of graupel and not from the total content of graupel". How large is this difference and why does it make such a difference?

**Response**: In the scheme the graupel is produced by the snow collecting liquid water, then the graupel growths by collecting other species (and with vapour deposition). If the graupel collects liquid species (rain or cloud), then there are two possibilities: the graupel is able to freeze the liquid content collected (this is called the dry growth mode) or not and a thin liquid film appears at the surface graupel (this is called the wet growth mode). The choice is made with the help of a heat budget.

> *the graupel growths mainly by collecting other species. When this collection implies liquid species (rain and/or cloud), there are two possibilities (called growth mode) depending on a heat balance: the graupel is able to freeze the entire collected liquid collected (dry mode) or a thin liquid film appears at the graupel surface (wet mode). In the original version of the ICE scheme there was confusion between the maximum content of liquid water than can be frozen (which must be used in the heat balance) and the content of liquid and ice water that can be collected in wet growth mode (which must be used to compute the graupel tendency). The correction made the mode choice more continuous. And because, ultimately, the graupel growth mode has an impact on the collection efficiency of icy species (snow and cloud ice) with the graupel, this choice can lead to significant differences in the collection rates. Hence, the scheme including the correction is less time-step dependent;*

If hail is activated, the wet growth mode contributes to the formation of hail. A conversion rate from graupel to hail is computed based on the collection rates and on the heat budget used to choose the growth mode. Then there are two ways of using this conversion rate:

- the old one: the conversion rate is applied to the entire graupel content. A given percentage of the graupel is then transformed into hail. For a same simulation length, depending on the number of time steps used, this conversion is made a different number of times. To simplify the reasoning, if the conversion rate is constant and equal to 0.5, the percentage of graupel converted into hail after 10s is 50% with one time step of 10s and 75% with two time steps of 5s.

- the proposed one: the conversion rate is only applied to the tendency of the graupel. The graupel already present at the beginning of the time step remains graupel and only the collected mass (in wet mode) can be converted into hail. This modification reduces the amount of hail produced and suppresses the time-step dependency.

The proposed version must still be validated but it is better than the old one because the time-step dependency is suppressed and because the old version had a tendency to produce small amount of hail under nearly all precipitating cold clouds (and this is no longer the case with the new version).

> *several modifications have been carried out on the processes involving the hail category as a prognostic field: the processes dealing with hail are now completely symmetric with those dealing with the graupel category (to ensure consistency even if this did not produce time-*

*step dependency). A conversion fraction is computed from the heat balance used to choose the graupel growth mode. In the original version of the scheme, this fraction was applied on the total content of graupel; this induced a conversion tendency directly linked to the number of times the rate is applied (hence to the time step for a given simulation length). On the contrary, in the new version, the conversion fraction is applied on the wet growth rate, this way, no time-step dependency is produced. This was the main reason for the time-step dependency on the hail category.*

**Comment 9**: Figures 11, 12 & 13. please make clear that the x-axis is timestep length (dt) – it could also be interpreted as timestep number (i.e. as a time-height plot)

**Response**: Thank you for the suggestion, it is done.

**Comment**: Technical/language corrections:

- page 2, line 9-10. please provide mode details about what differences were seen in the Meso-NH model when the time step was changed?

- page 3, line 9. what is a "tool package". Where can the reader find it?

- page 3, line 13. "makes possible" → "makes it possible"

- page 4, line 1 correct to "consists of a python package written to ..."

- page 4, line 5 delete "which is the required"

- page 6, line 3 "needed to use the parameterization" → "needed by the parameterization"

- page 6, line 6 "that not exist" → "that do not exist"

-page 7, line 4 "to plug other" → "to plug in other"

- page 7, line 5 "one have to define" → "one has to define"

- page 8, line 2 "dependency on the simulation" → "dependency of the simulation"

- page 8, line 11 please quantify what you mean by "small time steps"- page 8, line 18 it is not clear to me what you mean by "graupel growth mode", please give more details

- page 8, line 23 "has melt into rain" → "has melted into rain"

- page 8, line 25 "insure" → "ensure"

-page 9, line 10 & 11. please use standard scientific notation (e.g. $1.0 \times 10^{-5}$ )

- page 9, line 11. please quantify the "substantial additional cost"

- page 9, line 16 "unique" → "single"

- repeated grammatical error (e.g. page 12, line 9; page 12, line 31; page 13, line 1, page 13, line 7): "Longer is the time step, more this part is important" → "The longer the time step is, the more important this part is"

- page 12, line 31. What is the "it" in "it reaches around 11%"?

- page 13, line 5 "mean content is weaker" → "mean content is less"

- page 13, line 20 "by consequences" → "as a consequence"

- page 13, line 24 "this induces" → "this means"

- page 13, line 25 "larger to one" → "larger than one"

- page 14, line 3 "None of both schemes" → "Neither of these two schemes"

- page 14, line 9 "weaker" → "less"

- page 14, line 28 "whatever is the time step" → "whatever the time step is"

- page 14, line 32 "hypothesis done" → "hypothesis"

-page 15, line 2 "certainly reduce" → "certainly reduced"

- page 15, line 10 "This scheme allows to make fall the bigger drops quicker" → "This scheme allows the bigger drops to fall more quickly"

**Response**: Thank you for the numerous corrections you have suggested. In addition, the paper have been reviewed by a native writer of English. Some of your remarks need a reply, they are below:

- page 2, line 9-10. please provide mode details about what differences were seen in the Meso-NH model when the time step was changed?

  ○ The test simulations have been done several years ago and are no more available. A rerun of those simulations would be necessary to give extended details on the differences.

- page 3, line 9. what is a "tool package". Where can the reader find it?

  ○ *A documentation is provided with the software (see the code availability section).*

- page 8, line 11 please quantify what you mean by "small time steps"- page 8, line 18 it is not clear to me what you mean by "graupel growth mode", please give more details

  ○ small time steps: replaced by *simulations using the small time steps (shown in Fig. 7)*

  ○ graupel mode: please see comment #8

- page 9, line 10 & 11. please use standard scientific notation (e.g. $1.0 \times 10^{-5}$ )

  ○ It was an error, the values are 0.01 and 0.05

- page 9, line 11. please quantify the "substantial additional cost"

  ○ *The iterations needed for the 0.05g kg$^{-1}$ threshold induce an cost increase of about 5%, and for the 0.01g kg$^{-1}$ threshold an additional 20% can be expected (this last figure is an estimation because no sensitivity test have been performed on the whole domain).*

- page 12, line 31. What is the "it" in "it reaches around 11%"?

  ○ sentence reformulated: *For the 60s time step, 11% of the total water reached the ground.*

[revised manuscript text omitted]

---

## Author Response (AR2)

Dear editor,

Thank you for your comments. I took them into account and the manuscript has been read again; I hope that the unclear statements are now corrected.

Some of your comments require a specific response (new manuscript text is italicized in the reply):

- P1, L14: How can a modification be hidden or amplified? Please reformulate.

  Following text has been added: "*through compensatory errors and feedbacks*"

- P2, L11: Which are the microphysical "core" processes?

  Sorry, they were defined by mistake on the second occurrence, I moved the definition to the first occurrence: "*collections, riming, vapor deposition, evaporation*"

- P2, L17: What exactly is "more realistic"? Is it still unrealistic or realistic?

  The time-step dependency still exists for realistic initial values. The word "more" is suppressed.

- P7, section 2.1.2: please check the whole section. They are numerous language issues which make the text difficult to understand. For instance, what means "among these options is the time step one"? Is this time step number one? Or is the time step one of the options? Should "once BY simulation" rather mean "once PER simulation? And what means "to surround this call with conversions"? How can I surround a call with something? and what kind of conversions do you mean here?

  The time step is one of the options.
  With "surround", I just meant before and after. The new sentence is: "*It might be necessary to perform numerical and/or physical conversions before and after this call.*".
  Examples of conversions are given in the following sentence.

- P8, L12/13: please reformulate the sentence in brackets. It is not clear what you want to say.

  As the sentence in brackets is not essential, I prefer to remove it rather than to hinder understanding.

- P8, L14: What is the "intensity" of variables? Is this simply their value?

  Yes.

[revised manuscript text omitted]

---

## Author Response (AR3)

Dear editor,

Thank you again for your comments. I have taken into account your suggestions and you will find below the modifications made to the manuscript (in italics).

• P2, Sentence beginning in line 26 must be rewritten. It is spoiled and thus unclear.

The sentence has been split in tow:

For example, after 60 s of simulation, great uncertainty exists on the hydrometeors presence; depending on the time step used, rain, graupel and snow may (with very significant content) or may not exist. It should be noted here that 60 s is the order of magnitude of the time-step length used in the Météo-France small scale operational numerical weather model, AROME (Application of Research to Operations at Mesoscale, Seity et al., 2011), which shares the same physical package with the Meso-NH model.

• P10, line 12: You state that the curves in Figures 8 and 9 are "perfectly indistinguishable". For Figure 9 I agree, but not for Figure 8. Please clarify.

The sentence is rewritten:

With the revised version of the microphysical scheme, the simulated values for the different simulations shown in Fig. 7 and Fig. 1 are now perfectly indistinguishable in Fig. 9 and Fig. 8 for every common output times (the simulation with a 60 s time step, with the dashed line in the figure, provides outputs only after 60, 120 and 180 s of integration time).

• P 12, Line 4: has this "other" scheme a name? Please be more concrete.

The scheme has no specific name and is called 'Eulerian scheme' in the manuscript. I worked on the sentence to suppress the word 'other' and explicit better where the scheme comes from.

Two schemes are available: the AROME operational one (Bouteloup et al. (2010), BSB2010 hereafter), which is a statistical scheme and the Eulerian scheme included in the original version of the ICE scheme.

• Line 9. change "important" to something else or delete it. I would also be happy if you work again on this sentence. I cannot understand how a hypothesis (something in our minds) can cause a problem in a computer code.

Maybe "The assumption made in the scheme" would have been better than "hypothesis". I simplified the sentence:

These schemes take into account a terminal fall speed directly linked to the mean hydrometeor content (the more the content, the more rapid the fall). This feature makes it difficult, for these schemes, to accurately resolve the sedimentation process.

• Line 19/20: I assume that "model schemes" does here refer to the two schemes that you are going to test. It would be good if you clarify this, because it is easy to think that you mean your two reference schemes.

The two schemes in use in the model (BSB2010 and the Eulerian scheme) however (...)

• Line 29/30: This is difficult to understand. I think I know what you mean but it takes a while to get it. It seems that you have used 600 different timesteps: 0.1, 0.2, .... 59.9, 60.0 seconds. The problem is that the word step is used twice in the sentence with completely different

meaning. Please reformulate, and do this as well in the figure caption of Figure 11.

I hope the new wording is easier to understand:

The top panel of Fig. 11 shows the resulting profiles for different time steps (600 simulations are performed using time steps between 0.1 s and 60 s with an increment of 0.1 s) using the BSB2010 scheme (this is not a time evolution, all profiles are the result of a 400 s long integration).

The caption of Figure 11 is changed accordingly:

Vertical profiles of the rain mixing-ratio (the color scale represents the mixing-ratio in g kg-1) after a 400 s long integration for different time steps (600 simulations are performed using time steps between 0.1 s and 60 s with an increment of 0.1 s, abscissa) for the BSB2010 scheme (upper panel) and the Eulerian scheme as available in the operational source code (lower panel).

• Page 13, first sentence: It is unclear what you want to say.

Phrase is rewritten:

The BSB2010 scheme behaves differently regarding the CFL number. For CFL numbers larger than one, the diffusion on the vertical is more intense than for CFL numbers smaller than one. And the result obtained for a number of one is completely different from the results obtained with other values.

• Page 14, lines 29, 30: The sentence in brackets needs reformulation. I cannot understand it.

The sentence is replaced by the following ones:

In addition, it will be necessary to overcome the time-step dependency issue in order to allow the comparison of the different schemes. One way to do this could be to make a compromise by selecting a time step that is not too large to be able to mask the timestep dependency and, at the same time, not too small to limit the computation cost.